# Organelle calcium-derived voltage oscillations in pacemaker neurons drive the motor program for food-seeking behavior in *Aplysia*

**Alexis Bédécarrats, Laura Puygrenier, John Castro O'Byrne, Quentin Lade, John Simmers, Romuald Nargeot\***

Univ. Bordeaux, INCIA, UMR 5287, F-33076 Bordeaux, Bordeaux, France

**Abstract** The expression of motivated behaviors depends on both external and internally arising neural stimuli, yet the intrinsic releasing mechanisms for such variably occurring behaviors remain elusive. In isolated nervous system preparations of *Aplysia*, we have found that irregularly expressed cycles of motor output underlying food-seeking behavior arise from regular membrane potential oscillations of varying magnitude in an identified pair of interneurons (B63) in the bilateral buccal ganglia. This rhythmic signal, which is specific to the B63 cells, is generated by organelle-derived intracellular calcium fluxes that activate voltage-independent plasma membrane channels. The resulting voltage oscillation spreads throughout a subset of gap junction-coupled buccal network neurons and by triggering plateau potential-mediated bursts in B63, can initiate motor output driving food-seeking action. Thus, an atypical neuronal pacemaker mechanism, based on rhythmic intracellular calcium store release and intercellular propagation, can act as an autonomous intrinsic releaser for the occurrence of a motivated behavior.

**\*For correspondence:**
romuald.nargeot@u-bordeaux.fr

**Competing interests:** The authors declare that no competing interests exist.

## Introduction

Motivated behaviors, such as feeding or sexual activity, are triggered by an interplay between impulsive signals originating within the central nervous system (CNS), peripheral stimuli such as sensory cues, and the positive or negative consequences of an act (*Balleine, 2019*; *Berridge, 2019*; *Berridge, 2004*; *Dickinson and Balleine, 1994*; *Fujimoto et al., 2019*). This combination of flexible extrinsic and intrinsic neural releasers determines both the likelihood of occurrence and the selection of action patterns, which in turn imparts irregularity to the expressed goal-directed behavior. However, depending on sensory experience motivated behaviors can be transformed from variable to regular, rhythmically repeating action patterns that lead to the expression of habits, routines, or compulsive behaviors. The production of such stereotyped repetitive behavior, often reinforced by associative learning processes, is considered to become more strongly dependent upon an automatic internally arising drive and less sensitive to the sensory consequences of the executed action (*Balleine, 2019*; *Balleine and Dezfouli, 2019*; *Everitt and Robbins, 2016*; *Everitt and Robbins, 2005*). Although the contribution of internal drives to the induction of motivated behavior is recognized, unanswered questions remain about their neural origin and whether the highly flexible expression of a motivated behavior relies on similar inherent neuronal processes as found for rhythmic behaviors generally (*Grillner and El Manira, 2020*; *Marder et al., 2015*; *Selverston, 2010*; *Steuer and Guertin, 2019*).

A suitable animal model for addressing such issues is the sea slug *Aplysia*, in which aspects of feeding behavior are generated by a well-characterized neuronal network within the buccal ganglia. In the absence of food stimuli, *Aplysia* spontaneously expresses food-seeking behavior, which in

addition to locomotor and head-waving movements, includes buccal and radula (a tongue-like organ) biting movements emitted at highly irregular intervals (*Kupfermann, 1974*). This spontaneous and variable behavior can be regulated by operant-reward conditioning that leads to the expression of regular and rhythmic biting movements (*Brembs et al., 2002*; *Costa et al., 2020*; *Nargeot et al., 2007*; *Sieling et al., 2014*). Importantly, neural correlates of this motivated behavior continue to be expressed by the underlying neuronal network in the isolated buccal ganglia, thereby enabling the mechanisms responsible for autonomously driving both the irregular and regular emissions of radula movement cycles to be analyzed at the cellular and synaptic levels (*McManus et al., 2019*; *Nargeot and Simmers, 2012*). Identified components of this central pattern generator (CPG) circuit, such as the electrically coupled B63, B30, B31/32 neurons, were previously found to be essential contributors to the decision-making process that drives radula motor output (*Costa et al., 2020*; *Hurwitz et al., 1997*; *Jing et al., 2004*; *Nargeot et al., 2009*; *Sieling et al., 2014*; *Susswein and Byrne, 1988*). Among these elements, the two bilateral B63 interneurons are the only cells whose spontaneous production of an action potential burst is necessary and sufficient to trigger each radula output cycle (*Nargeot et al., 2009*). Thus, deciphering the mechanisms underlying the bursting activity of these key decision neurons is critical to understanding the process of radula motor pattern expression. Although earlier modeling evidence suggested that B63 bursting might rely on the cell's electrical synapses with other circuit neurons that possess a plateau potential-generating capability (*Susswein et al., 2002*), the actual triggering process for spontaneous B63 bursts and consequently the irregular emission of buccal CPG output remains unknown. Our findings reported here indicate that such motor pattern genesis relies on a voltage-insensitive pacemaker mechanism that at least partly derives from organelle-driven fluxes in intracellular calcium in this pair of neurons themselves.

## Results

Motor output responsible for radula biting behavior, which in the absence of any food stimulation consists of irregularly recurring cycles of radula protraction, closure and retraction (*Figure 1A*), continues to be expressed by identified CPG circuitry in isolated buccal ganglia (*Figure 1B*) and can be recorded from the corresponding buccal motor nerves (*Figure 1C*). Individual radula bites are instigated by synchronous impulse burst activity in the two bilateral, electrically coupled B63 interneurons that via electrical and chemical synapses with their ipsi- and contralateral buccal network partners, are able to trigger the two-phase buccal motor pattern (BMP) for a bite cycle (*Figure 1C*; also see *Hurwitz et al., 1997*; *Nargeot et al., 2007*; *Nargeot et al., 2009*). This essential role played by B63 is partly mediated by sustained, large amplitude membrane depolarizations that activate high-frequency bursts of action potentials (*Figure 1C*; also see *Nargeot et al., 2009*). Consistent with these underlying depolarizations arising from a bistable membrane property (*Russell and Hartline, 1978*), a brief intracellular injection of depolarizing current into an otherwise silent B63 neuron can trigger a depolarizing plateau and accompanying burst discharge that far outlasts the initiating stimulus (*Figure 1D*). The stimulated B63 in turn activates a similar burst-generating depolarization in the contralateral B63 cell and elicits a single BMP by the buccal CPG network.

### A rhythmic oscillatory drive underlies irregular BMP genesis

To investigate the mechanism(s) responsible for spontaneously instigating the B63 neuron's plateau-like potentials and resulting BMPs, we first sought evidence for an underlying triggering process in stable intracellular recordings from this neuron in still active isolated buccal ganglia (N = 26) in the absence of any electrical or chemical stimulation. Such recordings (episodes of >10 min per cell) revealed that B63's membrane potential underwent continuous depolarizing fluctuations over time (*Figures 1C* and *2A*), many of which remained below threshold for action and plateau potential generation. Others of these low-amplitude depolarizations elicited isolated action potentials without a plateau depolarization, whereas the remainders were associated with the production of a plateau potential and the expression of a BMP. Consequently, B63's plateau potentials and fictive bite cycles were spontaneously generated at irregular time intervals ranging from tens of seconds to several minutes.

Although B63's widely variable plateauing activity was expressed in an apparently random manner, we next asked whether its recurrence was associated with a specific temporal organization in the cell's membrane potential fluctuations. To assess this possibility, Fourier (spectral) analysis (see

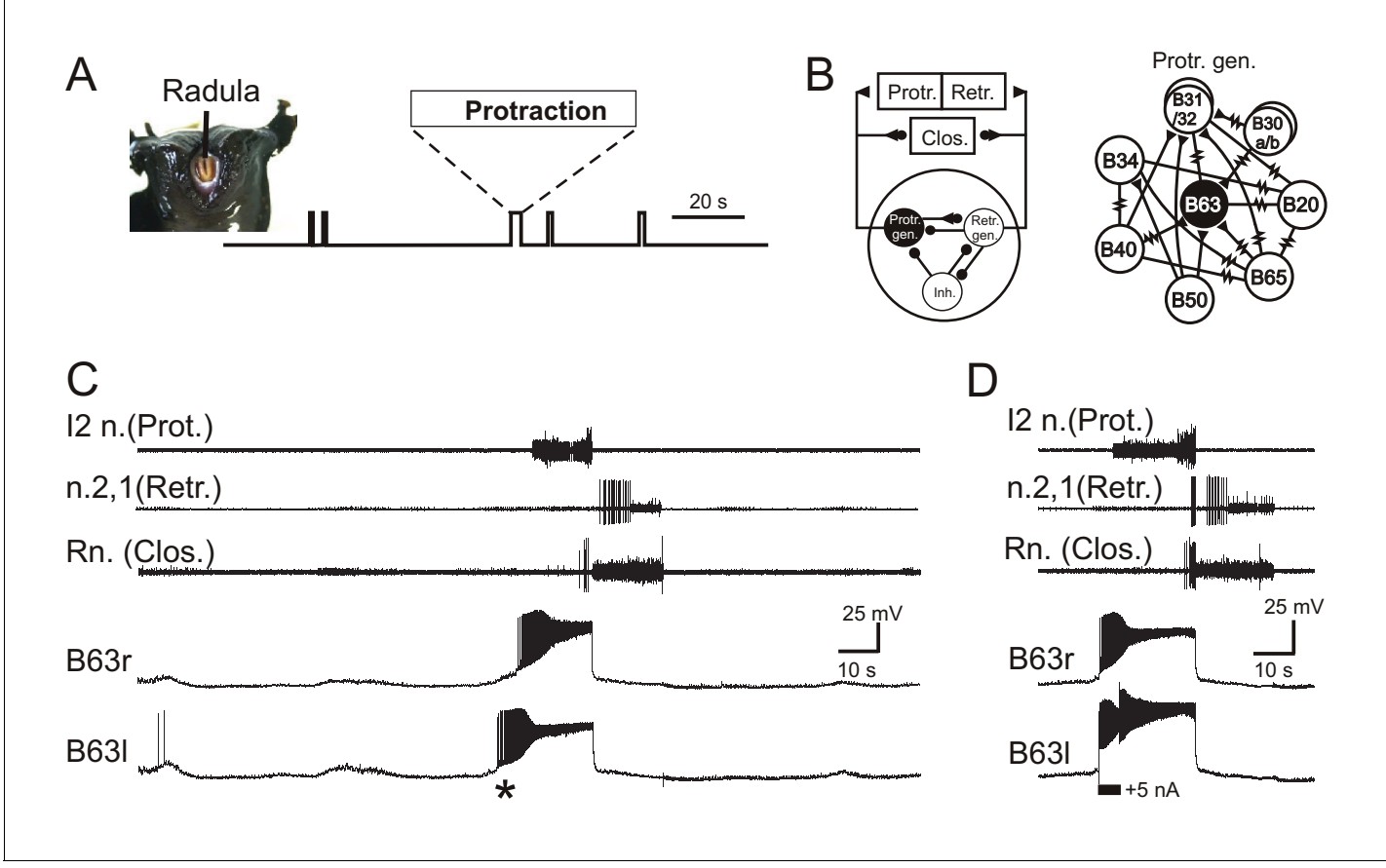

**Figure 1.** *Aplysia*'s spontaneous radula biting behavior and underlying motor pattern generation. (**A**) In vivo food-seeking behavior. In the absence of any external stimulation, *Aplysia's* radula (see head frontal view at left) spontaneously produces biting movements consisting of irregularly timed cycles of radula protraction (upward deflection of movement monitor trace), closure and retraction (downward deflection). (**B**) Schematics of the buccal CPG network that generates radula biting movements. Left: simplified diagram of the half-center network (one in each of the bilateral buccal ganglia) and its synaptic connections with protraction (Protr.), retraction (Retr.) and closure (Clos.) motoneurons (filled circles and triangles; inhibitory and excitatory connections, respectively). The network producing each bi-phasic cycle of movement is composed of three distinct and synaptically connected neuronal subsets comprising a protraction generator (Protr. gen.), a retraction generator (Retr. gen.) and a group of inhibitory neurons (Inh.). Right: detailed schematic of identified neurons belonging to the protraction generator and their synaptic interconnections (filled triangles, excitatory chemical synapses; resistance symbols, electrical synapses). Within the protraction generator, the neuron B63 (black) is necessary and sufficient to trigger the buccal motor pattern (BMP) for a radula bite cycle. (**C**) Simultaneous extracellular recordings of a single BMP (top three traces) and intracellular recordings of the two bilateral and electrically-coupled B63 neurons (r, right; l, left) in an isolated in vitro buccal ganglia preparation. I2n., n.2,1, Rn., are respectively the motor nerves carrying axons of protractor, retractor and closure motor neurons. The two B63 cells expressed spontaneous and coincident membrane depolarizations (*) that initiated plateau potentials and associated impulse bursts, which in turn evoked a BMP by the buccal CPG network. (**D**) Synchronous plateau potentials in the electrically coupled B63 and a resulting BMP triggered by a brief intracellular depolarizing current pulse (+5 nA) injected into one (left) B63 neuron.

Materials and methods) was applied to 10 min excerpts of the 26 B63 cell recordings. As seen in the spectral density periodogram (*Figure 2B*) for the B63 neuron illustrated in *Figure 2A*, the cell's spontaneous membrane potential changes decomposed into two distinct periodicities with peaks at 61 s and 144 s, respectively (*Figure 2B*, upper panel; *Figure 2—figure supplement 1A*). Moreover, a mathematical reconstruction based on these dominant periods showed that the slower waveform was correlated with the largest plateau depolarizations and the production of BMPs (*Figure 2C*). In contrast, the faster waveform was timed with virtually all membrane voltage changes, including the subthreshold fluctuations and events associated with isolated action potentials or plateau potential-driven bursts.

The faster of the two periodicities (mean ± CI95, 58 ± 5 s) varied relatively little between different preparations, as evidenced by the sharper spectral density peak in the averaged periodogram for all

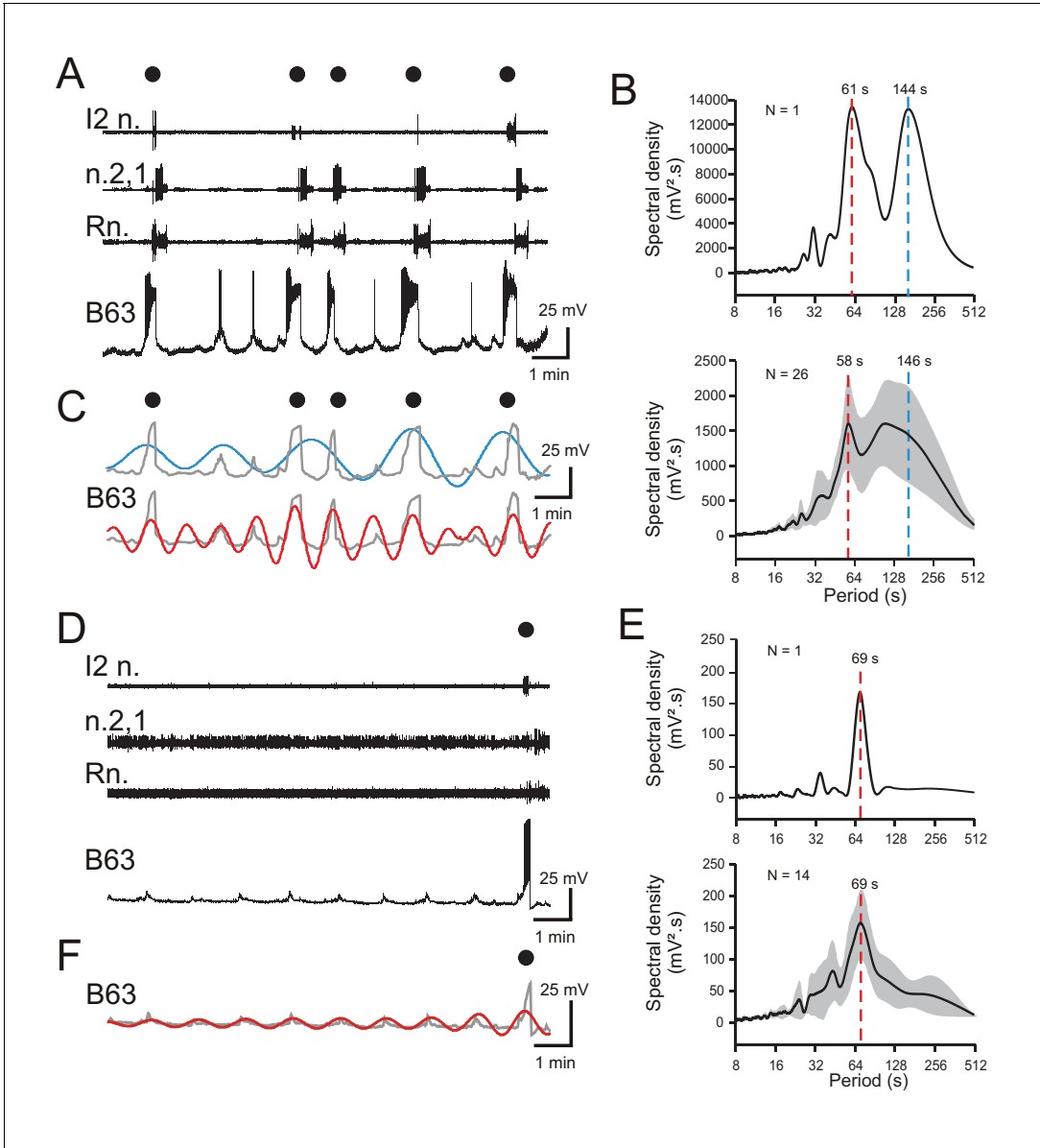

**Figure 2.** Periodicities in B63's spontaneous bioelectrical behavior. (**A**) A 10 min recording excerpt of radula BMP genesis (black dots) in an in vitro buccal ganglion preparation showing associated spontaneous fluctuations in membrane potential of an intracellularly recorded B63 neuron. Note that a BMP occurred only when B63 expressed prolonged burst firing driven by a plateau potential. (**B**) Spectral density plot of the B63 recording illustrated in A (top) and the average power spectrum (± CI95%) from recordings of 26 different neurons (bottom). In both cases, the essentially bimodal periodograms indicated that the variations in B63's membrane potential comprised two distinct periodicities (red and blue dashed lines indicate the means of these two dominant periods rather than peaks of average power), which across all 26 neurons was 58 s and 146 s, respectively. For details see *Figure 2—figure supplement 1A*. (**C**) Wavelet-based reconstructions retaining the two dominant periods revealed in the individual power spectrum in B (top) and their superposition with the smoothed membrane voltage traces (gray) of the corresponding B63 neuron in A. The slower sinusoid (blue trace; period, 144 s) corresponded to the cell's strongest depolarizations associated exclusively with the expression of plateau potentials and resultant BMP genesis (black dots). The faster sinusoid (red trace; period, 61 s) corresponded to these supra-threshold depolarizations plus almost all remaining subthreshold depolarizations. (**D–F**) Equivalent analyses of the same neurons as in A-C, but during recorded excerpts when no plateauing and BMP genesis occurred (N = 14). The single plateau potential and BMP occurring at the end of the excerpt in D is illustrated for comparison with the B63 recording in A, but was not included in the spectral analyses of E (see *Figure 2—figure supplement 1B*). In the absence of plateau potentials, the cells expressed spontaneous variations in membrane potential (**D**) composed of a single dominant, low-amplitude oscillation (**E,F**). Note that smaller additional peaks in the power spectra in **B,E** are essentially harmonics of the major period(s).

The online version of this article includes the following source data and figure supplement(s) for figure 2:

**Source data 1.** Spectral density plots of B63 membrane potential in ASW.

**Figure supplement 1.** Spectral density analysis of B63 neuron intracellular recordings.

26 buccal preparations (*Figure 2B*, lower panel; *Figure 2—source data 1*). In contrast, the broader peak of the slower rhythm (mean ± CI95, 146 ± 23 s) was indicative of the wide variability in occurrence of plateau potentials over time and between preparations. Moreover, for both rhythms, the considerable variability in their power spectral magnitudes (1441 ± 765 mV$^2$.s, 1524 ± 670 mV$^2$.s; mean ± CI95%, respectively) was attributable to the large amplitude variations between spontaneous membrane depolarizations that succeeded or failed to trigger plateau potentials in the different preparations (also see *Figure 2—figure supplement 1A*).

To further characterize the temporal nature of B63's faster oscillatory rhythm, spectral analysis was performed on cells (N = 14) that did not produce plateau potentials and resultant BMPs throughout 10 min recording sequences. Such non-plateauing neurons continued to express repetitive, now uniquely sub-threshold, membrane depolarizations (*Figure 2D*) that again were clearly rhythmic as revealed by the single dominant peak both in individual (*Figure 2E*, top) and averaged periodograms of the 14 recorded neurons (*Figure 2E*, bottom; *Figure 2—source data 1*). Moreover, the mean period (± CI95%) of this solitary rhythm (69 ± 7 s) was within a range equivalent to that of the faster oscillatory waveform found in B63 neurons that additionally expressed plateau potentials (compare *Figure 2E and B*). Although the magnitude of the remaining subthreshold rhythm varied over time and between preparations (*Figure 2E*; *Figure 2—figure supplement 1B*), as also evidenced by waveform reconstruction (*Figure 2F*), it had a much smaller amplitude than the corresponding waveform in plateau-expressing cells.

Altogether, these results show that B63's spontaneous bioelectrical behavior includes a rhythmic depolarizing signal that can remain below threshold for neuronal excitability or, in an apparently random manner, can lead to action and plateau potential production. However, despite their irregularity, the expression of plateau potentials, and resultant BMPs, is also inscribed with a periodicity, albeit considerably slower and more variable than the underlying oscillation.

## The voltage oscillation is endogenous to B63 and drives plateauing

In principle, the low amplitude oscillation of the B63 neurons could originate extrinsically from a presynaptic source, or derive intrinsically from a rhythmogenic property inherent to the neurons itself. Although no other buccal ganglia cell has been found to provide such a synaptic drive, we distinguished between these two possibilities by recording B63 cells in isolated preparations in which chemical synapses were blocked by bath perfusion of a modified saline containing a low calcium concentration (3 mM) and 10 mM cobalt, a nonspecific calcium channel blocker.

The application of such 'Low Ca+Co' saline soon induced a prolonged depolarization of recorded B63 neurons (*Figure 3A*), then after ca. 20 min, which was necessary to fully block chemical synapses - as confirmed by the suppression of the excitatory synapse between B63 and a contralateral B31 neuron (data not shown) - the membrane potential repolarized to its initial level. Significantly, these neurons thereafter continued spontaneously to express a low-amplitude oscillation for > 1.5 hr, although its magnitude gradually decreased over time (see *Figure 9—figure supplement 1A*). As in normal saline conditions, the cyclic depolarizations were either sub- or supra-threshold for spike generation, or at irregular intervals, were associated with sustained plateau-like depolarizations and high-frequency bursts (*Figure 3A,B*; *Figure 3—figure supplement 1*). Significantly, the continued expression of this burst-generating capability under functional synaptic isolation confirmed that the underlying plateau potentials, as suggested by evidence reported above (see *Figure 1C*), arose from an endogenous membrane property of the B63 neurons themselves. It also is noteworthy that in Low Ca+Co saline, due to the resultant suppression of burst-terminating inhibitory synaptic input from buccal neurons of the retraction generator (see *Figure 1B*), B63's plateau potentials still had variable durations, but overall lasted longer than in normal saline (compare *Figure 2A* and *Figure 6—figure supplement 1A*).

Spectral analysis of 20 min recording periods after chemical synapse blockade in 15 buccal ganglia preparations revealed that, as in ASW conditions, B63's membrane potential fluctuations decomposed into two major periodicities (*Figure 3B,C*; also see *Figure 2—figure supplement 1C*; *Figure 3—source data 1*). The periodograms and corresponding waveform reconstructions indicated that the slowest oscillation (mean period ± CI95%: 274 ± 61 s) of large magnitude (mean spectral density ± CI95%: 10214 ± 3764 mV$^2$.s) was mainly associated with the expression of plateau potentials. The fastest oscillation (mean period ± CI95%: 104 ± 12 s) of smaller amplitude (mean spectral density ± CI95%: 4679 ± 2699 mV$^2$.s) corresponded to rhythmic depolarizations that

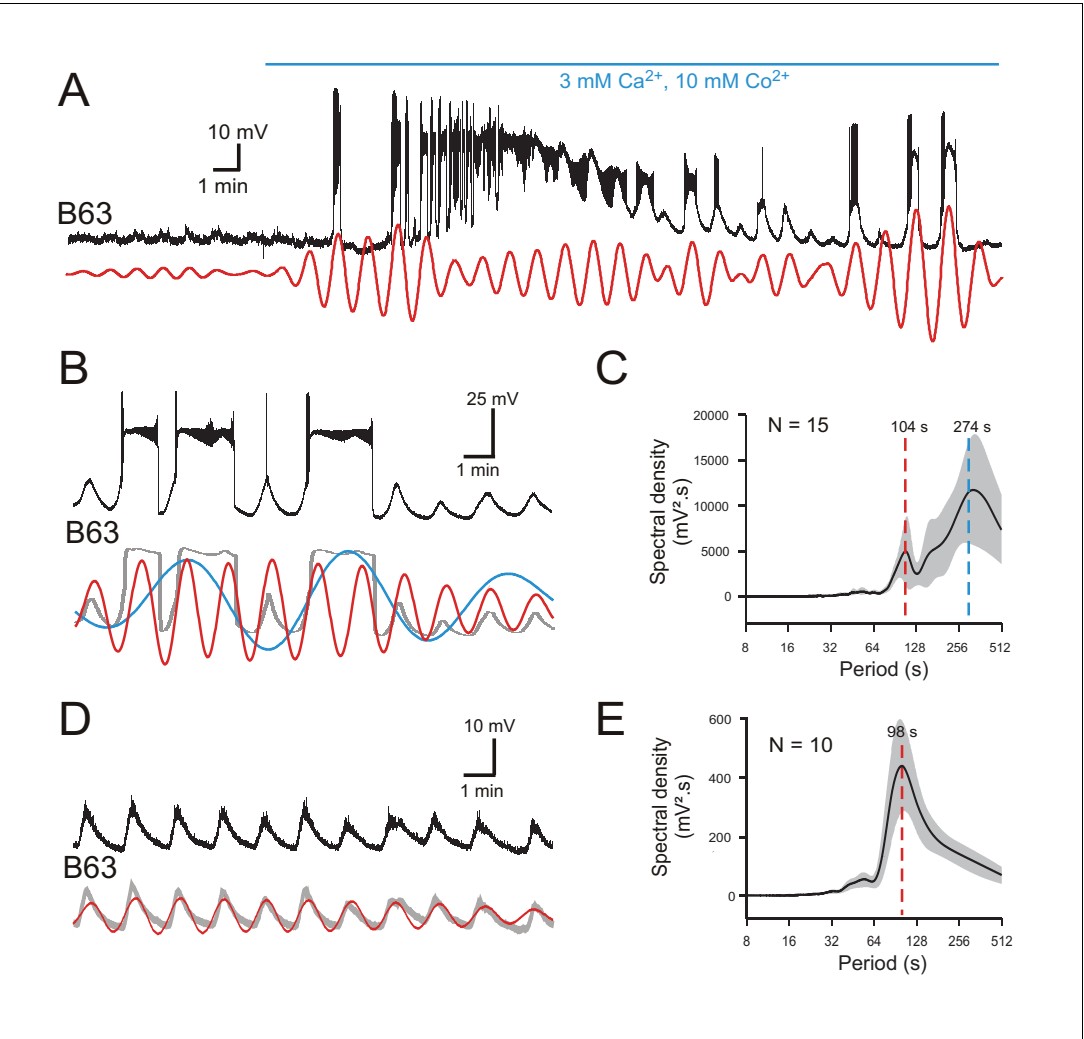

**Figure 3.** B63's voltage oscillations in the absence of functional chemical synapses. (A) Membrane potential fluctuations and plateauing in a recorded B63 neuron immediately before and during onset of bath-applied Low Ca+Co saline (horizontal blue line) to block chemical synapses in the buccal CPG network. Red trace: corresponding reconstructed waveform from the peak spectral density (period range: 70–90 s) (B,C) A different B63 cell recorded 20 min after onset of Low Ca+Co perfusion (B, Top trace; also see *Figure 2—figure supplement 1C*). The membrane potential variations decomposed into two oscillatory waveforms (B, red and blue traces) with periods of 83 and 280 s, respectively. Gray trace: raw recording after smoothing. (C) Average power spectrum (mean period ± CI95%) from 15 neurons showing two major oscillations. (D,E) Same analysis as in B,C, but of B63 recording sequences without plateau potential generation (also see *Figure 2—figure supplement 1D*). The remaining spontaneous variations in membrane potential now comprised a single oscillation (D, red trace: period 85 s), as also indicated by the solitary dominant period in the averaged periodogram (mean ± CI95%) from 10 B63 neurons (E).

The online version of this article includes the following source data and figure supplement(s) for figure 3:

**Source data 1.** Spectral density plots of B63 membrane potential in Low Ca+Co saline.

**Figure supplement 1.** Expression of B63 plateau potentials with chemical synapses blocked Intracellular recording of a B63 neuron in a buccal ganglion under bath-applied Low Ca+Co saline.

remained subthreshold, or were associated either with low frequency spiking or plateau driven bursts. As found in unblocked ganglia, this faster periodicity was more clearly evident when plateauing was absent: in 10 of the 15 buccal ganglia, B63 failed to produce plateaus during at least 10 min of analyzed recording excerpts, although these neurons continued to spontaneously express a rhythmic subthreshold oscillation (*Figure 3D,E*; also see *Figure 2—figure supplement 1D*; *Figure 3—*

*source data 1*). Again, the mean cycle period of this solitary waveform (± CI95%: 98 ± 8 s) was similar to that of the faster rhythm when plateau potentials also occurred (compare *Figure 3C and E*).

Thus, although blocking chemical synapses led to variations in mean cycle periods, amplitudes and plateau durations, B63's spontaneous voltage fluctuations still expressed two distinct oscillatory states, indicating that both processes occur independently of chemical synaptic inputs. Furthermore, inspection of the superimposed reconstructions of these oscillations under Low Ca+Co (*Figure 3B*), as in ASW (see *Figure 2C*), indicated that the onset of each plateau potential was invariably associated with a depolarizing phase of the faster oscillation, suggesting that the latter membrane voltage signal might be responsible for triggering the former.

This initiating process was further indicated by comparing the kinetics of B63's spontaneous voltage changes during subthreshold cycles of oscillation with those associated with plateaus. From recordings under both synaptic blockade (*Figure 4A,B*) or normal saline conditions (*Figure 4—figure supplement 1A*), the superposition of single cycles with and without plateau potential occurrences indicated that the relatively fast rising phases of the two events shared similar trajectories. In the absence of a plateau potential, this initial depolarization could trigger large amplitude impulse firing, or when a plateau occurred, it emerged as an additional and sustained (lasting tens of seconds) depolarization of 20–30 mV that in turn elicited a high frequency burst of low amplitude action

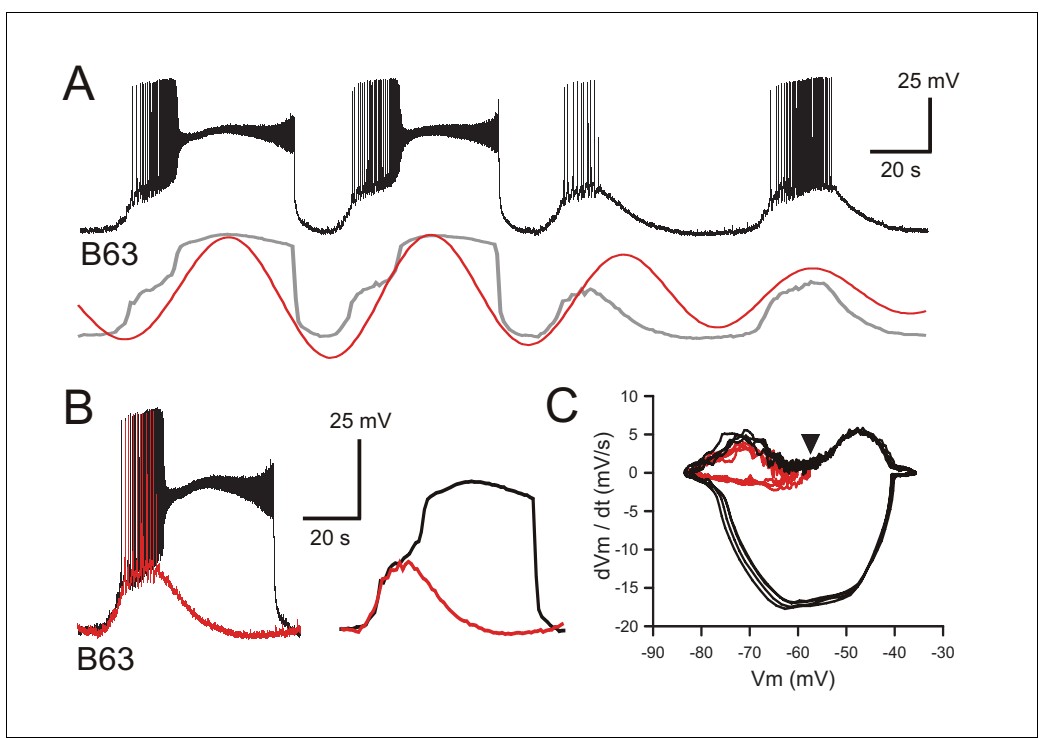

**Figure 4.** B63's voltage oscillation triggers plateau potentials. (**A**) Intracellular recording of a B63 neuron under chemical synapse blockade during an excerpt in which each spontaneous voltage oscillation was associated (first two cycles) or not (last two cycles) with the expression of plateau potentials. Bottom traces: corresponding smoothed recording (gray trace) and reconstructed oscillation from the peak spectral density (red trace; period 64 s). (**B**) Left: superposition of the first oscillation cycle in A with an accompanying plateau (black trace) and the third cycle without a plateau (red trace). Right: same traces after low-pass filtering to remove action potentials. (**C**) Phase-plane plot of 8 successive oscillation cycles both without (four cycles, red), and with (four cycles, black) plateau potential generation in the same B63 neuron as in **B, C**. The initial raising phases of the sub- and supra-threshold depolarizations follow identical trajectories before either a return to baseline potential or a further depolarization into a prolonged plateau. The arrowhead indicates the voltage threshold for plateau potential generation.
The online version of this article includes the following figure supplement(s) for figure 4:

**Figure supplement 1.** Triggering of plateau potentials by B63's voltage oscillation under normal saline (ASW) conditions.

potentials. The relationship between the voltage oscillation and plateauing in both Low Ca+Co and ASW conditions was quantified by phase plane analysis, which enables visualizing the voltage trajectory of neuronal oscillatory activity independent of time. To this end, recordings from B63 were low pass filtered to remove action potentials, then membrane voltage was plotted against the first derivative dV/dt, which is proportional to the net membrane ionic current (*Zhu et al., 2016*). Such phase-plane plots from data excerpts of the same neurons under synaptic blockade (*Figure 4C*) or unblocked conditions (*Figure 4—figure supplement 1B*) clearly showed a close coincidence between the early depolarizing trajectories of the spontaneous oscillation whether they developed (black spirals) or not (red spirals) into a sustained plateau. Subsequently, depending on the membrane potential reached at the end of this initial phase, the level of which varied considerably from one cycle to another, the trajectories bifurcated to give rise either to the large and stereotyped voltage changes of plateau potentials, or if subthreshold, immediately spiraled back to the baseline potential.

Therefore, together these results support the conclusion that rather than being instigated by chemically mediated synaptic inputs, the repeated expression of plateau potentials by the B63 neuron is a direct consequence of a spontaneous membrane voltage oscillation of irregular magnitude originating from within the cell itself.

## The oscillatory mechanism is not voltage-dependent

A classical diagnostic feature of endogenous neuronal oscillators, the inherent rhythmogenic capability which typically derives from voltage- and time-dependent membrane channels, is a sensitivity of cycle frequency and mode of firing to different levels of membrane polarization (*Bal et al., 1988*; *Canavier et al., 1991*; *Mathieu and Roberge, 1971*). We therefore tested the voltage-dependence of B63's oscillatory mechanism by manipulating the cell's membrane potential during intracellular recordings from buccal ganglia exposed to Low Ca+Co saline. As described earlier in this condition, B63 neurons continued spontaneously to generate a voltage oscillation that included both subthreshold depolarizations and less frequent plateau potentials with accompanying intense bursts of impulses (*Figure 5A*, left). As seen in *Figure 5A* (right), a continuous experimental hyperpolarization by intracellular current injection suppressed the expression of plateau potentials, but with no observable effect on the frequency of the underlying oscillation. The latter remained similar to that expressed before the imposed hyperpolarization where individual depolarizing cycles were strictly time-locked with the raising phase of each plateau potential (see arrowheads in *Figure 5A*, left). These findings were therefore in accordance with the all-or-none, voltage-sensitivity of B63's plateau potentials that are activated by the low-amplitude voltage oscillation. They also indicated that a change in B63's membrane potential, either in response to experimental manipulation or during the plateau potentials themselves, neither changed the small oscillation's cycle period nor caused phase-resetting (also see *Figure 6A*, *Figure 3—figure supplement 1*, *Figure 6—figure supplement 1A,B*).

This voltage-insensitivity of B63's low amplitude oscillation was further established by comparing the effects of the same imposed membrane potential changes in different preparations. Using two-electrode current-clamp in seven preparations, B63 was initially held at −70 mV, a potential that was subthreshold for plateau genesis, and subsequently further hyperpolarized to −80 mV. No significant change in oscillation cycle period resulted from this hyperpolarization (*Figure 5B,C*; V = 16, p = 0.799). Similarly, in nine preparations continuous depolarizing current injection that shifted B63's membrane potential from −70 mV to −30 mV also had no significant effect on the period of ongoing oscillation (*Figure 5D,E*; V = 28, p = 0.553). In contrast to the cycle period, however, in the same experiments the amplitude of B63's oscillation was found to increase (*Figure 5B,C*) or decrease (*Figure 5D,E*) according to the sign of injected current. Presumably, this was due to the membrane potential shifting relative to the reversal potential of the depolarizing inward currents producing the oscillation (see below). Finally, very similar observations were made from a different set of B63 neurons recorded in unstimulated buccal ganglia in ASW (data not shown), confirming that the cell's voltage-independent oscillation was a spontaneous emergent property regardless of whether the buccal network remained functionally reduced or intact.

Together, these results are consistent with an expected contribution of intrinsic, voltage-dependent channels to plateau potential genesis in the B63 neuron, and confirm that they are triggered by the underlying voltage oscillation. On the other hand, however, our data show that the mechanism

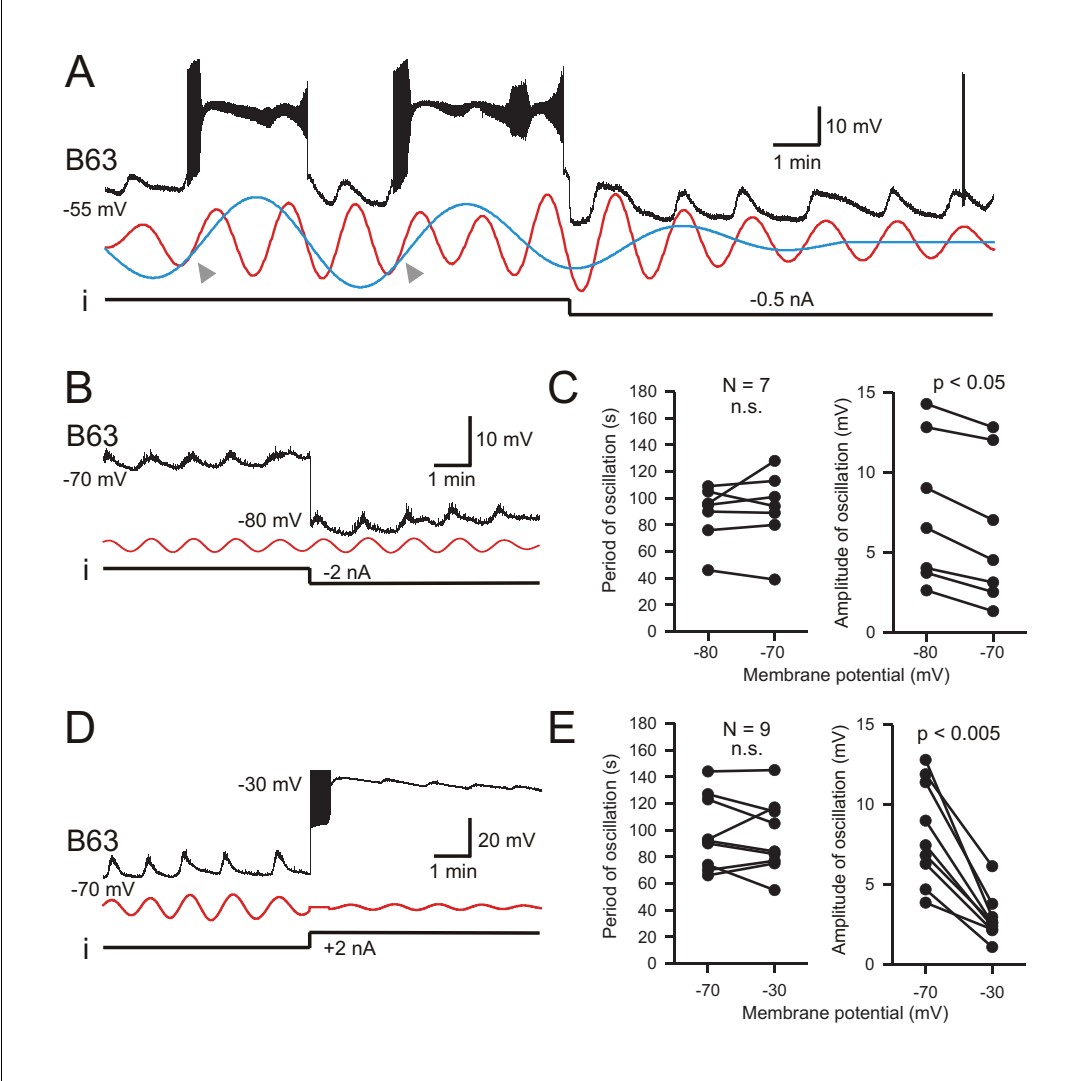

**Figure 5.** B63's low-amplitude oscillation does not arise from a voltage-sensitive mechanism. (**A**) Under chemical synapse blockade (with Low Ca+Co saline), a B63 neuron's spontaneous plateau potentials, but not its low-amplitude voltage oscillation, are suppressed by continuous hyperpolarizing current injection (i, −0.5 nA). Red and blue traces: superimposed reconstructed waveforms from the peak spectral densities corresponding to the presence or absence of plateau potentials. Arrowheads indicate the points of waveform intersection where plateau potentials were initiated. (**B**) Low-amplitude oscillation (upper trace) in a different B63 neuron during continuous hyperpolarization with chemical synapses blocked. The cell's membrane potential was held at −70 mV then stepped to −80 mV by continuous intracellular current injection (i) with two-electrode current clamp. Red trace: reconstructed waveform from the peak spectral density (period 80 s). (**C**) The oscillation cycle periods of all seven recorded neurons (left) were not significantly (n.s.) modified by the same membrane potential manipulation (V = 16, p = 0.799). However, this hyperpolarization significantly increased the oscillation amplitude (right, V = 0, p = 0.012). (**D,E**) Same analysis as in B,C, but with the membrane potential initially held at −70, then depolarized to −30 mV with two-electrode current clamp (**D**). Red trace: reconstructed waveform from the peak spectral density (period: 70 s). (**E**) No significant difference (n.s.) in oscillation period (left) was found in nine recorded neurons at these two holding potentials (V = 28, p = 0.553), whereas the depolarization caused a significant decrease in oscillation amplitude (right, V = 45, p = 0.0039).

responsible for the oscillation itself does not rely on an activation of voltage-dependent ion channels in the neuron's membrane.

## Circuit-wide voltage oscillation via gap-junction coupling

Although chemical synaptic interactions with other buccal network neurons are not responsible for generating B63's low-amplitude voltage oscillation, the possibility remained that it originates extrinsically and is conveyed to B63 through electrical synapses, which are widespread in buccal CPG circuitry. To assess this possibility, simultaneous intracellular recordings of B63 with at least one

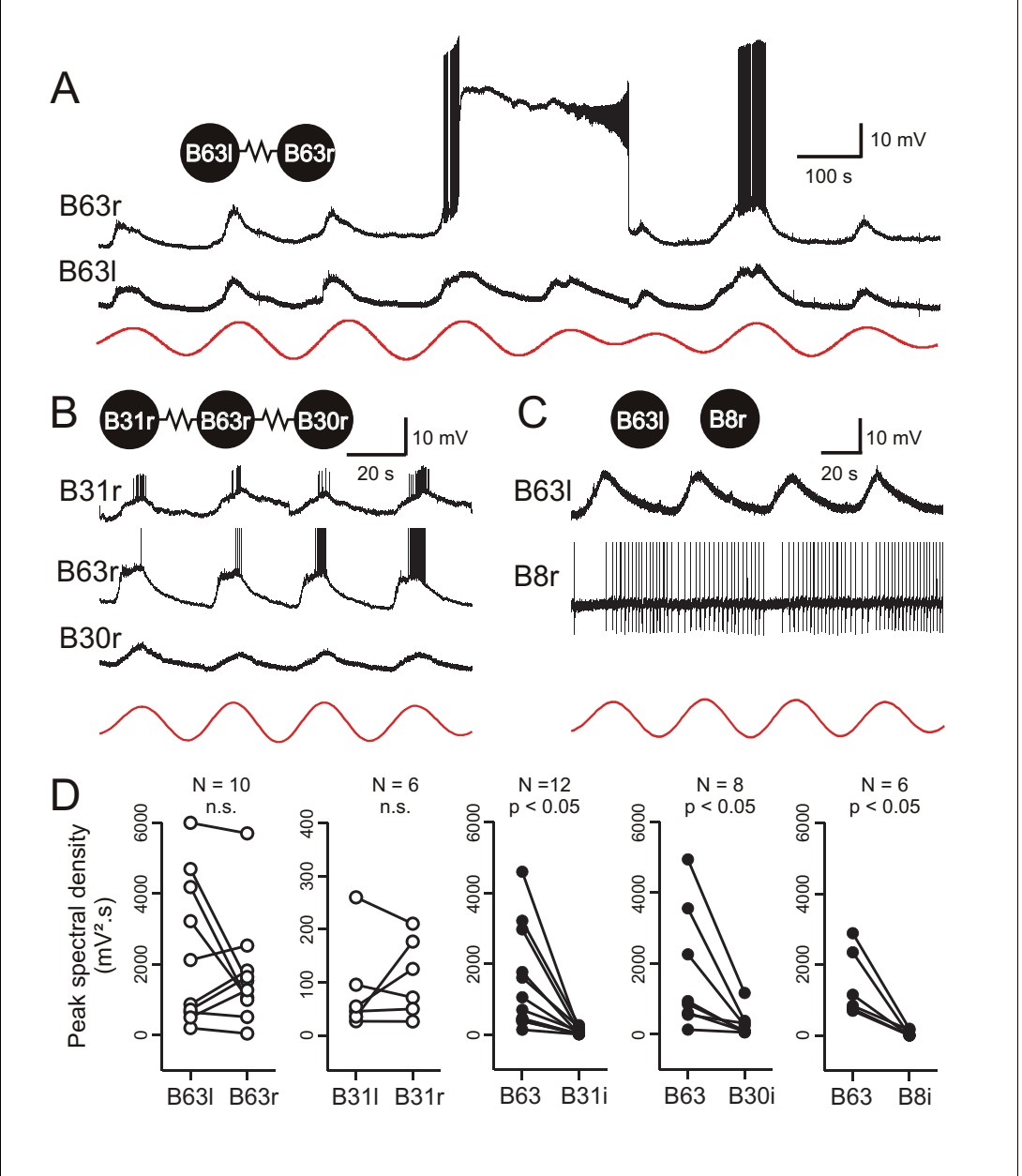

**Figure 6.** Low-amplitude oscillation in electrically-coupled network neurons. (**A,B**) Simultaneous intracellular recordings from different protraction generator neurons under chemical synapse blockade. (**A**) Spontaneous membrane potential oscillations in the right (r) and left (l) electrically coupled B63 cells (resistance symbol: electrical synapse). Note the independent expression of a plateau potential and burst firing in B63r. (**B**) Coordinated oscillations in a right B63 and ipsilateral, electrically coupled B31 and B30 neurons (the action potentials in the B63 trace are truncated). Red traces in A and B: reconstructed waveforms from the peak spectral densities for B63l and B63r, respectively. (**C**) Simultaneous intracellular recordings from a B63 cell and a non-coupled contralateral B8 motor neuron (action potentials in the B8 trace are truncated). A membrane voltage oscillation was absent in B8. Red trace: reconstructed waveform from the peak spectral density for B63. (**D**) Comparison of oscillation magnitude (i.e., peak spectral amplitude) in contralateral (unfilled dots; l, left; r, right) and ipsilateral neurons (i, filled dots). The oscillation amplitude was not significantly different (n.s.) in bilateral homologous cells (white dots; B63r/B63l, V = 22, p = 0.625; B31r/B31l, V = 13, p = 0.688), but was significantly higher in B63 compared to heterologous neurons in the same (i, ipsilateral) ganglion (black dots; B63/B31i, V = 78, p = 0.005; B63/B30i, V = 36, p = 0.008; B63/B8i, V = 21, p = 0.031).

The online version of this article includes the following figure supplement(s) for figure 6:

*Figure 6 continued on next page*

*Figure 6 continued*

**Figure supplement 1.** Irregular plateau potentials triggered by regular voltage oscillations in electrically coupled network neurons.

another electrically-coupled neuron of the buccal CPG network (see *Figure 1B*) were made under Low Ca+Co saline. Because the B63, B31, and B30 neurons in each of the bilateral ganglia are major components of the radula protraction generator subcircuit and share strong electrical synapses (*Hurwitz et al., 1997*; *Nargeot et al., 2007*) these three cell types were chosen for paired recordings. Other protraction generator neurons electrically coupled with B63, such as B34, B65, were also occasionally recorded, while B8 radula closure motor neurons, which are connected to these neurons via chemical, but not any electrical, synapses were used as a control (*Costa et al., 2020*).

Paired recordings from the bilateral B63 neurons, which are themselves electrically coupled, revealed that the two cells express almost identical low-amplitude oscillations that occur in strict synchrony (*Figure 6A*). However, action and plateau potentials, whose expression presumably depends on individual cell excitability, occurred independently. Moreover, within a same ganglion, B31 and B30 neurons belonging to the protraction generator and electrically coupled with the ipsilateral B63 also expressed a voltage oscillation in time with that of the latter (*Figure 6B*, *Figure 6—figure supplement 1A,B*). In contrast, B8 motor neurons, which are not coupled with B63 or the other protraction generator neurons, did not express any such oscillation (*Figure 6C*).

The amplitudes and phase relationships of the low-amplitude voltage oscillations in neuronal pairs were next quantified by spectral analysis over five successive cycles during which no plateauing occurred. The oscillation magnitude was determined from the peak spectral density of the single dominant period in the corresponding power spectrum (see *Figure 2—figure supplement 1D*). For identical and bilateral (homologous) neurons, no significant difference in oscillation amplitude was found between either the B63 or B31 cell pairs (*Figure 6D*, left; B63/B63, V = 22, p = 0.625; B31/B31, V = 13, p = 0.687). However, a comparison between different (heterologous) neuron pairs within a same ganglion showed that the oscillation magnitude was significantly greater in B63 than in either the ipsilateral B31 or B30 cells (B63/B31, V = 78, p < 0.001; B63/B30, V = 36, p < 0.01) and predictably, in B8 motor neurons (*Figure 6D*, middle to right).

Bivariate cross-waveform analysis of the same recordings revealed no significant phase difference in the voltage oscillations of homologous cell pairs, either between the two B63 or B31 neurons, in bilateral ganglia (*Figure 7A,C*; B63/B63, $V_0$ = 31, p = 0.769; B31/B31, $V_0$ = 15, p = 0.437). Unexpectedly, however, in heterologous ipsilateral pairs, B63's oscillation was found to be significantly phase-advanced by several seconds compared to the accompanying oscillation of either the B31 or B30 neurons (*Figure 7B,C*; B31/B63, $V_0$ = 78, p < 0.001; B30/B63, $V_0$ = 35, p < 0.02).

These findings thus showed that a spontaneous membrane potential oscillation is not restricted to the B63 neurons, but extends to all other neurons with which these two cells are electrically-coupled in the radula protraction generator circuit. The voltage oscillations in homologous cells in the two hemi-ganglia are synchronous and with similar amplitudes. However, within a given ganglion, each cycle of oscillation is expressed earlier in B63 and with a greater magnitude than in any of the cell's network partners.

## Involvement of cation channels and organelle signaling in B63's oscillation

As reported above, an experimental depolarization of B63 decreased the amplitude of its spontaneous voltage oscillation, indicating a reversal potential for the underlying ionic currents above −30 mV (see *Figure 5D*), which in turn suggested the involvement of sodium and/or calcium conductances in the oscillation. To test this likelihood, we examined two groups of six isolated buccal ganglia that were all initially bathed in Low Ca+Co saline to block chemical synapses. In a first group, the sodium channel blocker, TTX (50 μM), was then added to the bathing solution; in a second 'calcium-free' group, the initial saline was replaced by a solution lacking any calcium and containing the calcium chelator EGTA (0.5 mM).

As evidenced by the individual cell recordings in *Figure 8A and B*, the voltage oscillation of the B63 neurons was reversibly abolished by exposure to each of the salines. This suppression was

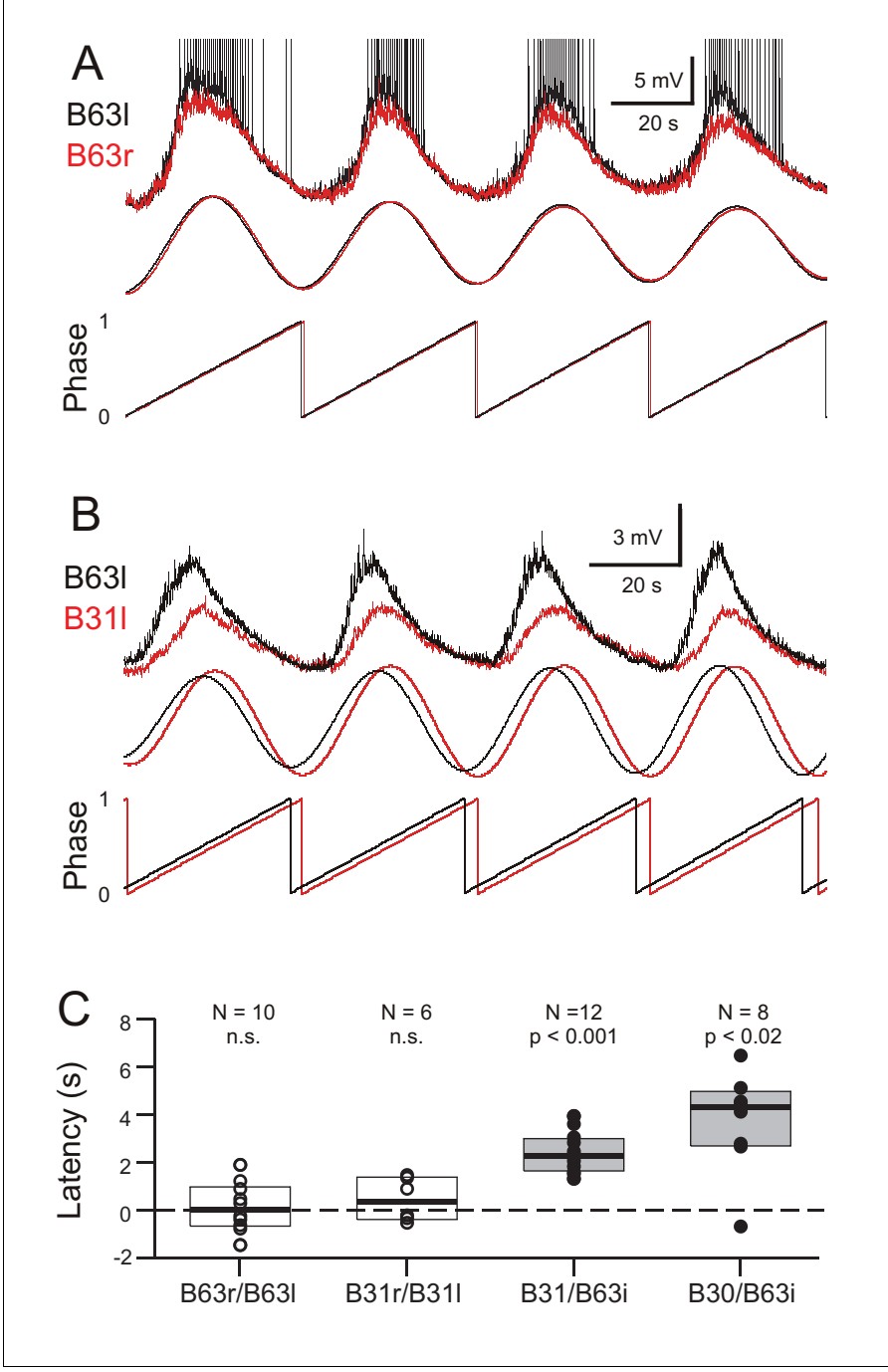

**Figure 7.** Phase-relationships between the oscillations of different network neurons. (A,B) Upper traces: Superimposed phase-aligned intracellular recordings from different neuronal pairs – (A), left (black) and right B63 (red); (B), left B63 (black) and left B31 (red) – under chemical synapse blockade (action potentials in B63l are truncated). Middle traces, reconstructed waveforms from the corresponding spectral periodograms after equivalence amplitude scaling. Lower traces: superimposed representations of the oscillation phases in each cell pair. No phase difference was evident between the two B63 neurons (A). In contrast, the oscillation of B63 (black) was phase-advanced relative to that of B31 (B). (C) One-sample analyses showing that the oscillation latencies in homologous bilateral neurons were not significantly (n.s.) different from zero (unfilled dots and boxes; B63r/B63l, $V_0 = 31$, $p = 0.770$; B31r/B31l, $V_0 = 15$, $p = 0.438$). In contrast, oscillations in heterologous neurons were significantly delayed (i.e. positive latency) relative to the ipsilateral (i) B63 partner (filled dots and boxes; B31/B63i, $V_0 = 78$, $p = 0.0005$; B30/B63i, $V_0 = 35$, $p = 0.016$).

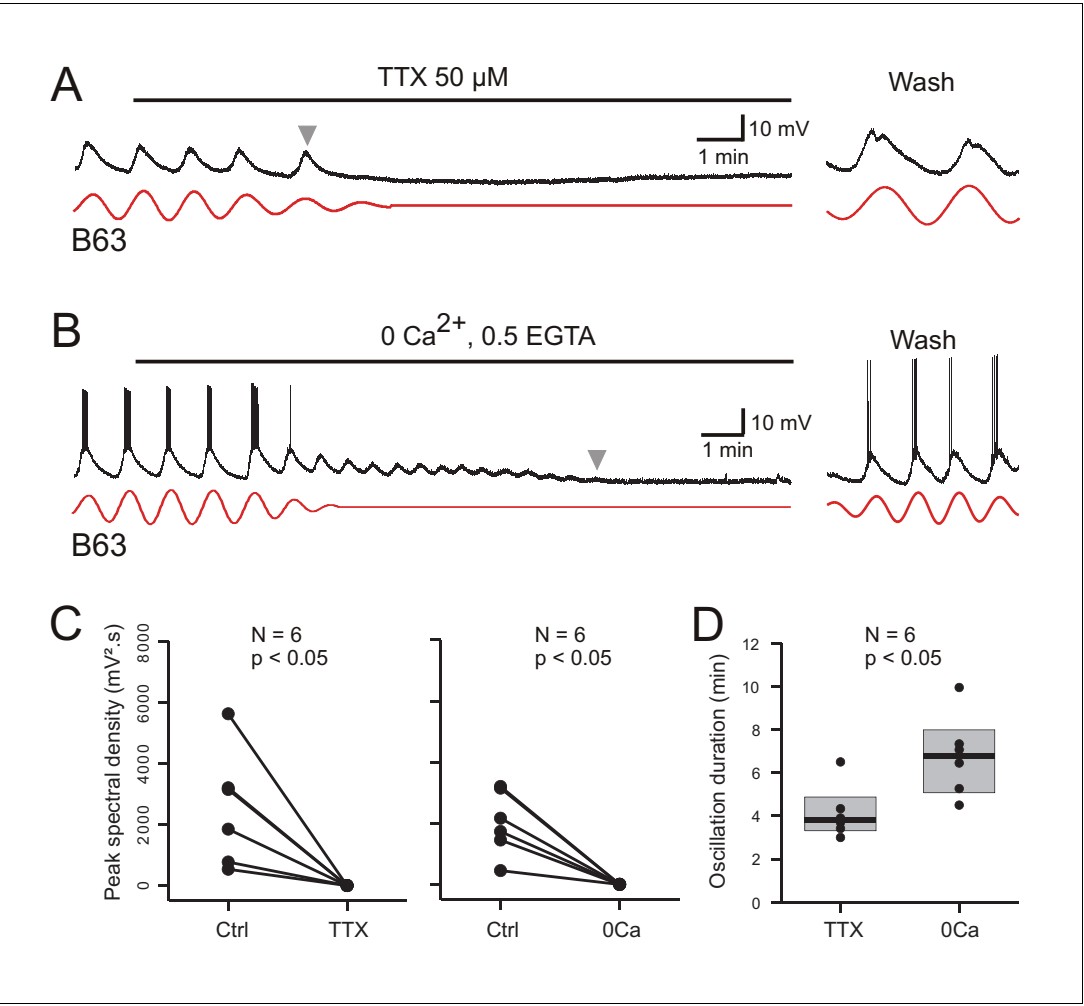

**Figure 8.** Involvement of sodium and calcium ions in the voltage oscillation. (A,B) Under chemical synapse blockade, the spontaneous voltage oscillations of B63 neurons were reversibly (trace excerpts at right) suppressed by bath application of 50 µM tetrodotoxin (A, horizontal line), or calcium-free salines (B, horizontal line). Red traces: reconstructed waveforms from corresponding peak spectral densities. (C) Group quantification under the experimental conditions illustrated in A,B: The amplitude of the dominant oscillation in Low Ca+Co saline alone (Control, Ctrl) was significantly reduced after application of TTX-containing (left, V = −21, p = 0.031), or calcium-free saline (right, V = −21, p = 0.031). (D) Inter-group comparison of oscillation longevity (gray arrowheads in A-C) after modified-saline perfusion onset (W = 3, p < 0.015). B63's oscillation persisted for significantly longer after removal of extracellular calcium (0 Ca + 0.5 EGTA) than after blockade of sodium channels by TTX.

quantified in the 12 recorded neurons by making a paired comparison of their peak spectral densities in 10 min data excerpts obtained before and after 10 min of modified saline application (*Figure 8C*). In all cases, the initial dominant oscillation was significantly diminished in each of the saline conditions (TTX: V = 21, p < 0.05; Calcium-free: V = 21, p < 0.05). A noticeable difference, however, was that from the instant when observable saline effects began to occur, the time course of this suppression varied considerably according to the saline condition. Whereas B63's oscillation terminated totally and abruptly in TTX-containing salines (*Figure 8A*), with the same rate of calcium-free perfusion, the oscillation persisted after an effect first became evident, damping slowly until its full suppression several minutes later (*Figure 8B*). This difference in oscillation longevity is further evident in the group analysis of *Figure 8D*, which compares the time until the oscillation ceased when measured from the onset of each modified saline's perfusion. Again, suppression took significantly longer in the Ca-free saline as compared to the TTX condition (W = 3, p = 0.015).

These results are therefore consistent with sodium and calcium ions playing a critical role in B63's spontaneous voltage oscillation, although their contributions appear to be fundamentally different. The rapid and full suppression of the oscillation in TTX-containing saline, which contained calcium, indicated that TTX-sensitive sodium channels are essential to producing the oscillation. By contrast, its slow decline in the absence of extracellular calcium is not consistent with a primary role of trans-membrane calcium influxes in oscillation genesis per se. Rather, although necessary for oscillation, calcium may act in an underlying regulatory process involving the dynamics of intracellular calcium and its control by intracellular stores, and that this signal is temporarily preserved after the cation's extracellular removal as the store calcium gradually runs down until depletion.

The main organelles that regulate intracellular calcium concentration are the endoplasmic reticulum (ER) whose membrane carries calcium channels, the calcium-ATPase reuptake pump (SERCA) and calcium release channels (the inositol triphosphate (IP3) and ryanodine (Ry) receptors), and mitochondria that act in energy supply as well as calcium sequestration and release (*Groten et al., 2013*). To test the implication of ER and mitochondrial calcium in B63's voltage oscillation, isolated buccal ganglia (N = 6) were bathed in Low Ca+Co saline before and after addition of 20 μM CPA, a selective inhibitor of SERCA (see Materials and methods). In a second group of ganglia, (N = 6), the same protocol was used, but with the addition of 20 μM FCCP, an oxidative phosphorylation uncoupling agent that leads to calcium release from mitochondrial stores. From intracellular recordings of B63 neurons in these preparations, peak spectral density magnitudes during a 10 min excerpt before drug application - and in the absence of plateau potentials - were compared to those computed over a 10 min period that began 20 min after the start of drug perfusion.

Bath perfusion of CPA caused a progressive and complete, but reversible, suppression of B63's voltage oscillation in association with a slight, but consistent, gradual membrane depolarization

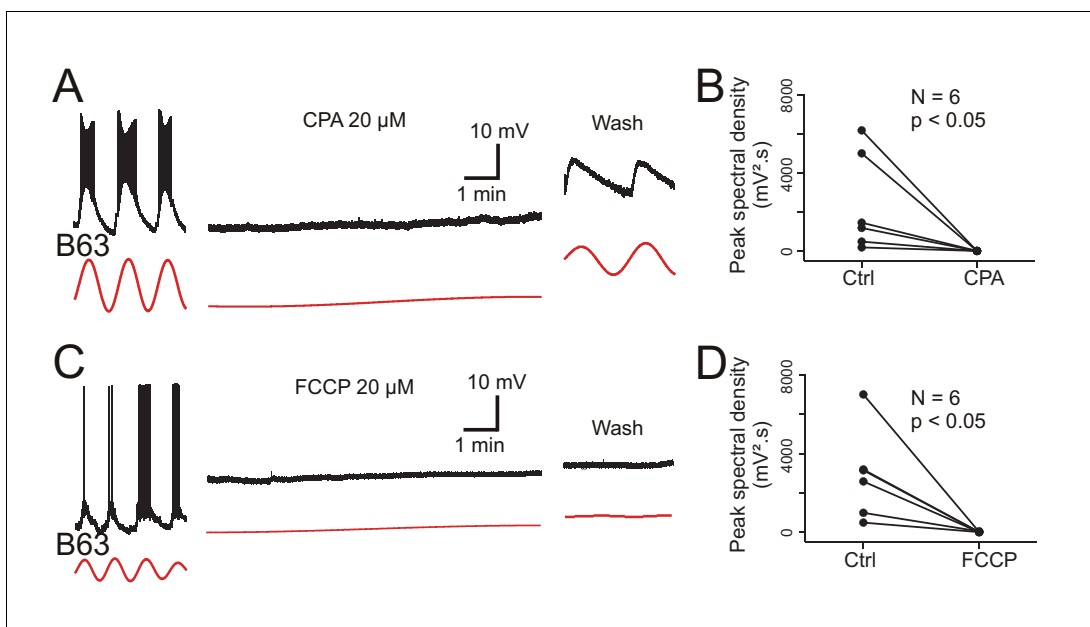

**Figure 9.** Involvement of intracellular calcium stores in the voltage oscillation. (**A**) The spontaneous voltage oscillation of a B63 neuron (left excerpt) was reversibly (right excerpt) suppressed in the presence of 20 μM CPA, a SERCA inhibitor (middle excerpt, recorded 20 min after the beginning of drug application). (**B**) Group analysis showing a significant reduction in oscillation magnitude of 6 B63 neurons measured before (Ctrl) and 20 min after the beginning of CPA application (V = 21, p = 0.031). (**C**) Suppression of B63 oscillation by application of 20 μM FCCP, an uncoupler of mitochondrial oxidative phosphorylation leading to calcium release. The neuron's spontaneous oscillation (left) was irreversibly (right) suppressed and the cell depolarized (middle, recorded 20 min after the beginning of drug application). (**D**) The oscillation magnitudes (Ctrl) of 6 tested B63 neurons were significantly reduced (V = 21, p = 0.031) after 20 min of FCCP application.

The online version of this article includes the following figure supplement(s) for figure 9:

**Figure supplement 1.** The voltage oscillation of B63 is unaffected by exposure to DMSO.

(*Figure 9A*). The application of FCCP also completely, although irreversibly, suppressed the oscillation that was now accompanied by a stronger sustained depolarization of ⌄10–20 mV (*Figure 9C*). No such change in B63's voltage oscillation or baseline membrane potential resulted from perfusion of either Low Ca+Co alone or this saline containing solely the DMSO vehicle (*Figure 9—figure supplement 1*). A within group analysis of peak spectral densities before vs. during drug application confirmed that exposure to CPA or FCCP significantly reduced the oscillation amplitude of all the recorded B63 neurons (*Figure 9B,D*; CPA: V = 21, p < 0.05; FCCP: V = 21, p < 0.05). The peak spectral density reduction was also significantly different between both the CPA and FCCP experimental groups and neurons exposed to DMSO alone, but not between the CPA and FCCP groups themselves (H = 11.415, p < 0.01; CPA vs. DMSO: q = 7.018, p < 0.001; FCCP vs. DMSO: q = 6.517, p < 0.001; CPA vs. FCCP: q = 0.501, p = 0.933).

These data are therefore consistent with the hypothesis that intracellular organelles play an important role in generating the B63 neuron's low-amplitude voltage oscillation by a dynamic regulation of intracellular calcium concentration via the release of store calcium and its sequestration mediated by ATP-dependent pumps. Depletion of mitochondrial calcium (induced by FCCP) or of ER calcium by an impairment of reuptake pumps (by CPA) would be expected to block this dynamic, leading to a rise in intracellular calcium levels and a resultant tonic cell membrane depolarization, which is precisely what we observed in the experiments reported above (see *Figure 9A,C*).

To further establish the ER's involvement in B63's voltage oscillation, a final series of experiments were conducted in which we assessed the effects of blocking the membrane calcium channels of the organelles themselves. This was achieved by pressure injecting heparin (20 mg/ml), a well-known non-permeable IP3 receptor antagonist (*Bezin et al., 2008*), into the somata of either bilateral pairs of B63 neurons, or their two B31 network partners. After 30 min injection, simultaneous intracellular recordings were made from heterologous B63 and B31 cell pairs under Low Ca+Co saline conditions. Heparin injection into the two B31 neurons had no effect on the ongoing voltage oscillation of either a heparin-injected B31 itself or its non-injected B63 partner (*Figure 10A*). In contrast, the reverse experiment that consisted of injecting the IP3 receptor antagonist into the two B63 neurons caused a drastic reduction in the spontaneous oscillation, both of one of the injected B63 cells and a recorded B31 partner (*Figure 10C*). These findings were further supported by spectral analysis of recording excerpts from B63 cells in the two groups of preparations after bilateral B31 (N = 4) or B63 (N = 5) heparin injection. Recorded B63 neurons continued to express a distinct voltage oscillation with a mean period of 67 ± 3.2 s when heparin was injected into the two B31 cells (*Figure 10B*, *Figure 10—source data 1*), but this dominant oscillation disappeared with heparin's presence in the B63 neurons (*Figure 10D*, *Figure 10—source data 1*). Therefore, in addition to the participation of ER calcium sequestering pumps, organelle calcium release via IP3-dependent calcium channels evidently contributes to the voltage oscillation of buccal CPG network neurons.

Significantly moreover, these results, along with the previous findings of a consistently larger amplitude and phase advance of B63's voltage oscillation (see *Figure 6*), further indicated that it originates in, and is specific to, the B63 cell pair - the sole necessary and sufficient elements for triggering BMPs - and spreads presumably via gap junctional connections throughout the remaining circuit.

## Discussion

This study aimed to decipher the basic neuronal mechanisms underlying a central network's ability to generate the impulsive motor drive for an aspect of *Aplysia's* food-seeking behavior. Our findings indicate that this highly irregular motivated act arises from an atypical pacemaker property of a homologous pair of decision-making interneurons belonging to the animal's buccal feeding network. The pacemaker signal does not derive from an oscillatory mechanism based on voltage-dependent ionic currents, but rather depends on a cyclic release/reuptake of calcium from intracellular stores acting on voltage-insensitive membrane channels. The resultant oscillation in membrane potential, which spreads to gap junction-coupled network partners, is blocked by an IP3 receptor antagonist injected selectively into the two decision neurons, indicating that the command process for the oscillation originates, at least predominantly, in these specific cells. The low amplitude oscillation is spontaneously expressed with a regular period but a varying magnitude. Depending on the membrane potential reached during the depolarizing phase of a given oscillation cycle, a prolonged plateau

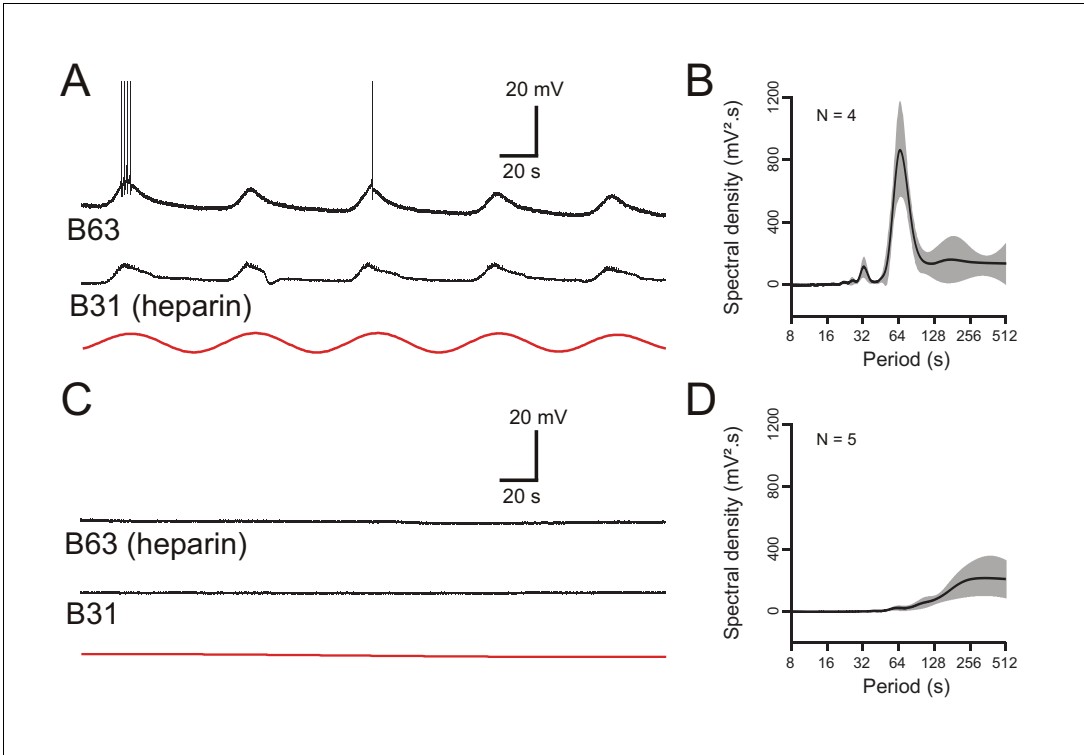

**Figure 10.** The voltage oscillation generated by intracellular calcium store release is specific to B63. (A,C) Paired recordings of B63 and B31 neurons under chemical synapse blockade, 30 min after the beginning of an intrasomatic injection of the ER membrane calcium channel blocker heparin (20 mg/ml) into either the bilateral B31 (A) or bilateral B63 (C) neurons. Heparin in B31 had no effect on the ongoing oscillation of an un-injected B63 cell (A), but suppressed oscillations in both a B63 and an un-injected B31 (C) after injection into both B63 neurons. Red traces: reconstructed waveforms from the corresponding periodograms. (B,D) Average power spectra obtained 30 min after the beginning of bilateral intracellular heparin injection into the B31 (B) or B63 neurons (D) in 4 and 5 preparations, respectively.

The online version of this article includes the following source data for figure 10:

**Source data 1.** Spectral density plots of B63 membrane potential after heparin injection into B63 or B31.

and accompanying spike burst may be initiated, which in turn elicits network output for a cycle of food-seeking movement. The calcium dynamic in two key circuit neurons thus provides a continuous rhythmic carrier signal from which burst-generating plateau potentials necessary for behavioral action can sporadically arise.

## Intracellular calcium oscillation as a neuronal pacemaker mechanism

The endogenous oscillatory capability of invertebrate and vertebrate neuronal pacemakers is mainly attributed to sets of plasma membrane ion channels whose specific functional properties allow the production of cyclic membrane depolarization/repolarization and associated impulse bursting (*Adams and Benson, 1985*; *Brocard et al., 2013*; *Calabrese, 1995*; *Chevalier et al., 2016*; *Golowasch et al., 2017*; *Harris-Warrick, 2002*; *Selverston, 2010*). Although this pacemaker mechanism can be regulated by second-messenger cascades and cytosolic calcium released from intracellular stores, its expression relies essentially on the voltage-sensitivity of the membrane channels themselves (*Butera et al., 1996*; *Canavier et al., 1991*; *Kadiri et al., 2011*; *Liu et al., 1998*; *Yu et al., 2004*). Thus, depending on membrane potential levels, the cycle frequency of the endogenous voltage oscillation can be modified, thereby changing the frequency of the effector rhythm in which the pacemaker cell is involved (*Canavier et al., 1991*; *Chevalier et al., 2016*; *Koshiya and Smith, 1999*; *Miller and Selverston, 1982*). Spontaneous neuronal oscillations can also be generated by voltage-independent pacemaker mechanisms involving plasma membrane ionic pumps, such as Na/K ATPase, which periodically repolarize the membrane of bursting neurons

(*Darbon et al., 2003*; *Johnson et al., 1992*; *Kueh et al., 2016*). Such pump-driven oscillations require tonic cellular activation or disinhibition and are not blocked by extracellular calcium removal or Na channel blockers such as TTX. Finally, non-excitable glial cells can contribute to the expression of rhythmic electrical activity in neighboring neurons via their regulatory effect on the surrounding ionic (including extracellular calcium) and chemical environment or by direct neuron-glia interactions through gap junctions (*Alvarez-Maubecin et al., 2000*; *Deitmer et al., 1998*; *Morquette et al., 2015*).

An increasing body of evidence from studies on endocrine, muscle and non-excitable tissues (*Baker et al., 2016*; *Fridlyand et al., 2010*; *Vinogradova et al., 2005*; *Zhou et al., 2019*), but also in early developing neurons (*Gu et al., 1994*), has indicated that a slow oscillatory cell signal with cycle periods of seconds to several minutes can be generated spontaneously by organelle-derived fluctuations in intracellular calcium concentration. Such a rhythmic calcium dynamic, involving notably the endoplasmic reticulum and mitochondria, may be a source of plasma membrane voltage oscillation without the participation of voltage-sensitive ion channels. Specifically, the oscillation arises from a periodic accumulation/removal of cytoplasmic calcium, principally by IP3 and Ry receptor-mediated calcium efflux and ATP-dependent pump-mediated influx across the store membrane, which in turn is translated into a voltage signal by an activation of calcium-sensitive channels at the plasma membrane (*Fridlyand et al., 2010*; *Hickey et al., 2010*; *van Helden et al., 2000*; *Vinogradova et al., 2005*). Glial cells can also express such intracellular calcium oscillations (*Deitmer et al., 1998*). Although in most cases these require specific inducing stimuli, such as neuronal activity, the presence of an excitatory transmitter or mechanical stimulation (*Charles et al., 1991*; *Cornell-Bell et al., 1990*; *Morquette et al., 2015*), spontaneous calcium oscillations in glia have been reported (*Wang et al., 2006*).

In a corresponding and novel manner for a neuronal system, our present data indicate that an intracellular calcium oscillation arising from organelle calcium release and reuptake in two specific neurons is responsible for the spontaneous low-amplitude voltage oscillation in these cells and their gap junction-coupled network partners: (1) the voltage oscillation originating in B63, which gives rise to a delayed and smaller oscillation in coupled neurons, is not only suppressed with pharmacological treatments that block SERCA pumps or disrupt energy production and calcium storage by mitochondria, but is also inhibited by the specific intracellular presence of an IP3 receptor antagonist in these two neurons; (2) experimental manipulation of the cell's membrane potential over voltage ranges where most voltage-dependent channels would be expected to be altered had no effect on either the occurrence or frequency of this oscillation, thus excluding a possible contribution of voltage-dependent channels to oscillation genesis. Nevertheless, such imposed voltage changes modified the amplitude of B63's ongoing oscillation, as expected by changes in ion fluxes through the manipulated cell's membrane channels; (3) at variance with a possible essential contribution of plasma membrane sodium or calcium pumps, the voltage oscillation was suppressed in TTX-containing saline and its magnitude increased, rather than decreased, as would be expected with low extracellular calcium concentrations. Presumably, the primary intracellular calcium dynamic in B63 neurons drives plasma membrane voltage oscillation by activating calcium-sensitive and voltage-insensitive sodium or other cation channels (*Hickey et al., 2010*; *Kadiri et al., 2011*; *Kramer and Zucker, 1985*).

The organelle calcium release and resulting voltage oscillation could be localized to a neuronal compartment, such as the junctional synapses or neuropile where a close proximity between ER, mitochondria, and plasma membrane is likely to exist (*Thompson et al., 1976*), but which is remote from the soma compartment where plateau potentials occur. In this scheme, local calcium release and IP3/calcium-dependent amplification/propagation along the ER membrane could activate nearby plasma membrane conductances at this soma distant site and generate a voltage oscillation that is passively backpropagated to the cell body where it can trigger plateau potentials. Such spatially separate loci for intracellular calcium oscillation and plateau potential production in turn provides a plausible explanation for several of our experimental observations, including: (1) the ability of local organelle calcium release to produce a membrane potential oscillation of B63, despite the presumed presence of strong cytoplasmic calcium buffering mechanisms; (2) the near synchronous expression of the voltage oscillation, but not the somatic plateau potentials, throughout B63's gap junction-coupled cell partners; and (3) the inability of experimental soma depolarization or

spontaneous plateau potentials and resulting somatic calcium influxes to modify or reset the remotely-produced calcium oscillation.

This organelle-driven oscillatory property of B63 does not exclude a possible contribution of a neuron-glia interplay in the generation of the voltage oscillation and/or its propagation through the buccal network, especially since neurons and glia can reciprocally modify their intercellular ionic environment or intracellular content via gap junctions (*Alvarez-Maubecin et al., 2000*; *Deitmer et al., 1998*; *Goldstein et al., 1982*; *Keicher et al., 1991*). Moreover, B63's membrane is known to carry burst-generating oscillatory properties other than those described in the present study (*Costa et al., 2020*; *Nargeot et al., 2009*; *Susswein et al., 2002*). Indeed, in contrast to the organelle-derived mechanism reported here, where B63 was behaving spontaneously in the absence of any experimental stimulation, this cell also possesses an oscillatory bursting capability that does rely on voltage-dependent ion channels (*Nargeot et al., 2007*; *Nargeot et al., 2009*; *Sieling et al., 2014*). However, this latter mechanism is activated only when the cell is conveyed to more depolarized levels by sensory-induced changes in excitability or in response to direct current injection. This state-dependent expression of two different burst-generating processes by the B63 neuron is therefore reminiscent of the multiple rhythmogenic ionic mechanisms reported in other oscillatory neurons, where each mechanism's participation varies according to different stimulus conditions (*Harris-Warrick and Flamm, 1987*; *Kadiri et al., 2011*; *Peña et al., 2004*).

## Variability in motor pattern emission with a periodic pacemaker mechanism

Irregularity in the expression of motor activity is a fundamental feature of motivated or goal-directed exploratory behaviors, including *Aplysia's* food-seeking movements, when animals are faced with an uncertain surrounding environment. Although such motor variability is partly dictated by peripheral sensory inputs (*Cullins et al., 2015*; *Lyttle et al., 2017*; *McManus et al., 2019*; *Pearson, 2000*; *Tam et al., 2020*; *Wimmer et al., 2015*), it also depends on the functional properties of the central networks and constituent neurons producing the behavior (*Sims et al., 2019*). In this context, random processes such as stochastic variations in the activation of intrinsic and voltage-dependent properties of individual neurons and synaptic noise can be sources of variability in motor output expression (*Carroll and Ramirez, 2013*; *Darshan et al., 2017*; *Melanson et al., 2017*; *Nargeot et al., 2009*; *Zhang et al., 2020*). Moreover, modeling evidence has suggested that an aperiodicity in slow cytosolic calcium dynamics can lead to irregular voltage oscillations in otherwise regularly bursting CPG neurons (*Falcke et al., 2000*). In contrast, our experimental data indicate that spontaneous and irregular motor pattern genesis can derive from a cell-specific pacemaker mechanism involving an intracellular calcium dynamic that itself is strictly periodic, but where randomness arises from cycle-to-cycle variations in the amplitude of the rhythmic membrane depolarizations it produces. By oscillating close to the threshold for voltage-dependent plateau potential genesis required for CPG circuit output, these low-amplitude depolarizations thereby determine the variability with which *Aplysia's* exploratory movements are expressed. Timing irregularity would be further reinforced by an interaction between the differing dynamics of the organelle-derived and voltage-dependent oscillatory mechanisms that coexist in the B63 neuron as mentioned above.

Magnitude alterations in cytosolic calcium fluxes and resultant plasma membrane voltage changes can arise from an interaction between different dynamic processes. Such variability could result from a direct interplay between the different intracellular calcium stores themselves (*Geiger and Magoski, 2008*; *Groten et al., 2013*; *Haberichter et al., 2001*; *Hajnóczky et al., 1995*; *Wacquier et al., 2019*; *Wacquier et al., 2016*) or from an interaction between the store-generated calcium oscillation and extracellular calcium influxes (*Chay, 1996a*; *Falcke et al., 2000*; *van Helden et al., 2000*). Furthermore, voltage amplitude irregularity could arise from an interplay between the individual calcium oscillations of gap junction-coupled neurons (*Bindschadler and Sneyd, 2001*; *De Blasio et al., 2004*; *Liu et al., 2011*). In addition to such processes, irregular magnitude fluctuations in the voltage oscillation of the B63 neurons could also partly result from plateau potential production in the different electrically coupled neurons of the buccal CPG network. Presumably, because these plateaus are generated in the soma, far from the intercellular junctions, they are not phase-coupled in the different network neurons, producing only weak depolarizations in post-junctional cell partners. Nevertheless, these uncoordinated plateau-related depolarizations in different

cells would be sufficient to participate in randomly modifying the amplitude of the ongoing voltage oscillation in the B63 neurons.

## Propagation of pacemaker activity amongst gap junction-connected neurons

It is well known that gap junction-mediated electrical coupling promotes the synchronization of pacemaker neuron bursting in CPG networks (*Leznik and Llinás, 2005*; *Marder, 1984*; *Nadim et al., 2017*; *Sasaki et al., 2013*; *Soto-Treviño et al., 2005*), and is similarly involved in *Aplysia's* buccal feeding circuit (*Sieling et al., 2014*). In non-neuronal tissues, gap junctions have also been found to co-ordinate multicellular activity by propagating calcium waves via metabolic coupling (*Benninger et al., 2008*; *Leybaert and Sanderson, 2012*; *Peters et al., 2007*; *Wang et al., 2006*). Due to strong intracellular buffering mechanisms, calcium itself is unlikely to play a role in such intercellular communication (*Leybaert and Sanderson, 2012*). Rather, calcium wave propagation through gap junctions is most likely mediated by a diffusion of IP3 and its chain activation of IP3/Ry receptors and calcium release within adjacent cells (*Harootunian et al., 1991*; *Miyazaki et al., 1992*; *Takeuchi et al., 2020*). In addition to transfer through gap junctions, calcium waves can be propagated by extracellular paracrine signaling involving calcium-induced transmitter release and an activation of membrane receptors and resultant IP3 synthesis in neighboring cells (*Newman and Zahs, 1997*; *Scemes and Giaume, 2006*).

Several lines of evidence suggest that the calcium dynamic driving membrane potential oscillation and originating in the B63 neurons is also conveyed non-electrically to its gap junction-coupled partners in the buccal network. First, the magnitude of B63's voltage oscillation, which is presumably proportional to the intracellular calcium signal, was consistently stronger than in any other network cells, such as B30, B31, and B65, despite their similar membrane input resistances. Second, the voltage oscillation in B63 preceded that recorded in these other cells by several seconds, which is compatible with a slower propagation (~70 μm/s) of the underlying calcium oscillatory signal by a metabolic process rather than by direct electrical transmission of the voltage oscillation itself (*Benninger et al., 2008*). Third, the intracellular injection of the IP3 receptor antagonist heparin into B63, but not into B31, suppressed the voltage oscillation in both neurons, thus indicating that its origin and intercellular propagation is selectively dependent on IP3 signaling in B63. These findings also argue against the possibility that the oscillation occurring throughout the buccal CPG circuit is a network property that emerges from electrical coupling between equivalently-active neurons, but rather, further underline the crucial pacemaker role played by B63 in buccal network operation. However, because our experiments were mainly conducted with all the network's chemical synapses blocked, we were unable to establish whether B63's endogenous oscillatory and plateau properties are alone sufficient in the decision process for BMP genesis. Nevertheless, in normal saline conditions with the network remaining functionally intact, in contrast to all other identified circuit cells, the B63 neuron pair are the only elements found to be necessary and sufficient for triggering motor pattern expression and resultant food-seeking movement (*Hurwitz et al., 1997*; *Nargeot et al., 2009*). Moreover, this essential leading role persists after appetitive operant conditioning-when the network's junctional conductances are strengthened and the transition from irregular to rhythmic BMP genesis occurs (*Nargeot et al., 2009*; *Nargeot and Simmers, 2012*).

In conclusion, without excluding the involvement of other cellular mechanisms, our study shows that in the absence of extrinsic stimulation, the CPG network output for *Aplysia's* food-seeking behavior can arise from a combination of spontaneous intracellular calcium dynamics in two decision neurons and IP3-dependent circuit-wide metabolic propagation. Although autonomously bursting neurons may employ intracellular stores as a source of calcium (*Kadiri et al., 2011*; *Levy, 1992*; *Scholz et al., 1988*), in all cases, the mobilization of store calcium, by interacting with calcium-activated membrane channels, is thought to regulate the voltage dynamics of ongoing bursting behavior. However, other than theoretical evidence (*Chay, 1996a*; *Chay, 1996b*), a spontaneous and rhythmic organelle-derived calcium dynamic serving as a primary oscillator mechanism for actually driving neuronal bursting has not been previously reported. Moreover, we believe that our findings provide the first example of the involvement of such a rhythmogenic mechanism in the highly variable expression of a motivated behavior. Experiments are now required to determine whether B63's intracellular calcium handling is regulated by associative learning when hungry *Aplysia* switches its

impulsive and irregular food-seeking movements to a rhythmic compulsive-like act as found in more complex organisms.

## Materials and methods

### Key resources table

| Reagent type (species) or resource | Designation | Source or reference | Identifiers | Additional information |
|---|---|---|---|---|
| Chemical compound, drug | Carbonyl cyanide 4-(trifluoromethoxy) phenylhydrazone (FCCP) | Merck-Sigma-Aldrich | C2920 | |
| Chemical compound, drug | Cyclopianozic acid (CPA) | Merck-Sigma-Aldrich | C1530 | |
| Chemical compound, drug | Dimethyl sulfoxide (DMSO) | Merck-Sigma-Aldrich | D5879 | |
| Chemical compound, drug | Ethylene glycol-bis (2-aminoethylether)-N, N, N ′, N′-tetraacetic acid (EGTA) | Merck-Sigma-Aldrich | E4378 | |
| Chemical compound, drug | Fast green | Merck-Sigma-Aldrich | F7252 | |
| Chemical compound, drug | Heparin sodium salt | Tocris | 2812 | |
| Chemical compound, drug | Tetrodotoxin (TTX) | Tocris | 1069 | |
| Software, algorithm | R | https://cran.r-project.org/ | | |
| Software, algorithm | PMCMRplus | https://cran.r-project.org/web/packages/PMCMRplus/index.html | | |
| Software, algorithm | WaveletComp | https://cran.r-project.org/web/packages/WaveletComp/index.html | | |

### Animals

Adult *Aplysia californica* (purchased from the University of Florida, Florida), and *A. fasciata* (caught locally in the Bassin d'Arcachon, France) were used in the experiments. Consistent with previous studies (*Katzoff et al., 2002*; *Sieling et al., 2014*), no inter-species differences in either behavioral or neuronal characteristics were found. Animals were housed in tanks containing fresh aerated sea water (~15°C) and were fed ad libitum with seaweed (*Ulva lactuca* obtained from the Station Biologique at Roscoff, *France*).

### Isolated nervous preparation

Animals were anesthetized with 50 ml isotonic $MgCl_2$ solution (in mM: 360 $MgCl_2$, 10 HEPES adjusted to pH 7.5) injected into the hemocoel. The bilateral buccal ganglia and their peripheral nerves were dissected from the animal and pinned out in a Sylgard-lined Petri dish containing a standard artificial sea water solution (ASW, in mM: 450 NaCl, 10 KCl, 30 $MgCl_2$, 20 $MgSO_4$, 10 $CaCl_2$, 10 HEPES with the pH adjusted to 7.5). The ganglia were desheathed to expose the neuronal somata and the preparations were continuously superfused with ASW at 15°C.

## In vitro electrophysiology

Spontaneous buccal motor output patterns were monitored by wire electrodes placed against appropriate motor nerves and insulated from the bath with petroleum jelly (Vaseline). The I2 (I2 n.), 2,1 (n. 2,1) and radular (R n.) nerves were used to monitor radular protraction, retraction, and closure activity, respectively (*Nargeot et al., 1997*). The motor pattern-initiating interneurons B63 and B30, and the motoneurons B31/B32 (protraction) and B8 (closure) were recorded and identified according to previously reported criteria (*Church and Lloyd, 1991*; *Hurwitz et al., 1997*; *Jelescu et al., 2013*; *Jing et al., 2004*; *Susswein and Byrne, 1988*). These neurons were impaled with sharp glass micro-electrodes with a tip resistance of 20–30 M$\Omega$ and filled with a $KCH_3CO_2$ solution (2 M). In the two-electrode current-clamp condition, two intrasomatic electrodes were inserted in each neuron, with one electrode used for current injection and the other for membrane potential recording via an Axo-clamp-2B amplifier (Molecular Devices, Palo Alto, CA). Intracellular and extracellular signals were digitalized and acquired at 5 kHz with a CED interface (CED 1401, Cambridge Electronic Design, UK) with Spike two software (Cambridge Electronic Design, UK).

## Modified saline and pharmacology

Blockade of chemical synaptic transmission was performed with bath perfusion of a modified ASW that contained cobalt, a calcium channel blocker ($CoCl_2$, 10 mM), and a lowered concentration of calcium ($CaCl_2$, 3 mM) (*Alkon and Grossman, 1978*). Neither this decrease in $Ca^{2+}$ concentration alone, nor the presence of the $CoCl_2$ alone was found sufficient to block the chemical synapses. This 'Low Ca+Co' saline contained (in mM): 446 NaCl, 10 KCl, 30 $MgCl_2$, 20 $MgSO_4$, 3 $CaCl_2$, 10 $CoCl_2$, 10 HEPES with the NaCl concentration adjusted to the same osmolarity as ASW. Synaptic blockade was indicated by the suppression of chemical excitatory post-synaptic potentials produced by B63 in the contralateral B31 neuron (*Hurwitz et al., 1997*). Data reported here under the Low Ca+Co saline condition were acquired after 20 min perfusion to allow for a complete synaptic blockade and the recovery of recorded neurons' resting membrane potential to at least −50 mV.

The calcium-free solution used in several experiments derived from the Low Ca+Co saline in which no calcium was present and a calcium chelator, Ethylene glycol-bis (2-aminoethylether)-N, N, N ', N'-tetraacetic acid (EGTA) was added (in mM: 450 NaCl, 10 KCl, 30 $MgCl_2$, 20 $MgSO_4$, 10 $CoCl_2$, 10 HEPES, 0.5 EGTA) (*Hickey et al., 2010*). The pH for all solutions was adjusted to 7.5.

Tetrodotoxin (TTX, Tocris), a blocker of sodium channels in plasma membranes, was diluted to 50 µM in distilled water from a 0.5 mM stock solution (*Hurwitz et al., 2008*). Cyclopianozic acid (CPA, Merck-Sigma-Aldrich), a blocker of the sarco/endoplasmic reticulum $Ca^{2+}$-ATPase pump (SERCA) and Carbonyl cyanide 4- (trifluoromethoxy) phenylhydrazone (FCCP, Merck-Sigma-Aldrich), a proto-nophoric uncoupler of mitochondrial oxidative phosphorylation that depolarizes the mitochondrial membrane and leads to the organelle's release of calcium, were diluted to 20 µM in Low Ca+Co saline from stock solutions that were prepared in dimethyl sulfoxide (DMSO) (*Benz and McLaughlin, 1983*; *Geiger and Magoski, 2008*; *Hickey et al., 2010*). The maximum concentration of DMSO in the final volume reached 0.05%, which in control and previously reported studies did not alter the electrophysiological properties of neurons, the strength of electrical synapses, or intracellular cal-cium concentrations (*Beekharry et al., 2018*).

Heparin sodium salt solution (Tocris) at 20 mg/ml, an inositol tri-phosphate (IP3) receptor antago-nist (*Bezin et al., 2008*), was pressure injected via a glass micropipette (10 M$\Omega$) inserted into the cell bodies of the bilateral B63 or B31 neurons. Pressure was generated by a Picospritzer2 with 20 pulses of 15 PSI, 150 ms, at around 0.03 Hz. Following injection, which was performed during bath perfu-sion of ASW, one of the two heparin-containing electrodes was removed and replaced by a 2 M KAcetate microelectrode for intracellular recording that started 30 min after the heparin injection. In several of these experiments (2/5 with B63 and 2/4 with B31), 2 mg/ml of fast green (Merck-Sigma-Aldrich) was added to the heparin solution to verify effectiveness of the injection. No difference was found between the intracellular recordings of cells injected with or without fast green.

## Intracellular recording analysis

Variations in the membrane potential of recorded neurons were analyzed in a cycle frequency/period bandwidth of 0.00195 to 0.125 Hz (i.e. periods of 512 s to 8 s) by Fast Fourier Transform (FFT) analy-sis using R language and environment (*R Development Core Team, 2019*) for statistical computing.

The membrane voltage recordings were initially smoothed using the Spike 2 'Smooth' filter with a time constant of 500 ms to suppress signals of frequencies higher than those within the desired band-width and down-sampled at 1 Hz in order to decrease computation time. The resulting power spectral density periodograms were then used to identify oscillation periods of peak magnitude (*Figure 2—figure supplement 1*). The periodograms were computed from the FFT frequency spectrograms by converting the frequency band (in Hz) to its reciprocal, period (in secs), to facilitate discerning the temporal correspondence between these plots and the relatively slow (from secs to mins) spontaneous membrane voltage fluctuations occurring in the raw recordings. Reconstruction of the sinusoidal waveforms corresponding to dominant periods and the phase-relationships between signals from neuron pairs were computed from Wavelet decomposition using the R-CRAN 'WaveletComp' package for the built-in analysis of univariate and bivariate time series (*Roesch and Schmidbauer, 2018*). Averaged periodograms represent means +/- 95% confidence interval (CI95%) of the individual periodograms. Phase-plane plots of membrane potential were computed by using custom-written R script (*Source code 1*) for intracellular recordings that were smoothed with a time constant of 500 ms and down-sampled at 10 Hz.

## Statistical analyses

Animals were randomly assigned to each experimental group, and although estimations of sample sizes were not computed initially, an attempt was made to minimize the number of animals sacrificed. One-sample comparisons to a theoretical value (0 s.) were performed using the two-tailed one-sample Wilcoxon signed rank test ($V_0$ statistic). Within-group comparisons of two datasets were carried out using the two-tailed Wilcoxon signed rank test (V statistic). Between-group comparisons of two independent groups were conducted using the two-tailed Mann-Whitney test (W statistic). The application of these non-parametric statistical tests to small datasets was justified by the failure to satisfy assumptions of normality and homoscedasticity with high statistical powers. Statistical analyses were performed using the R-CRAN 'Base' and 'PMCMRplus' packages (*Pohlert, 2019*). Similar results were obtained with analyses performed both with and without outlier values, and all statistics reported in the text and figures were computed without data removal. Differences were considered significant for $p < 0.05$. Box-plot illustrations represent median values (horizontal lines) along with the first and third quartiles (boxes) in datasets.

## Acknowledgements

This research was supported by grants ANImE ANR-13-BV5-0014-01 (ANImE, RN), ANR-10-Idex-03–02 (AB), and a doctoral studentship (LP) from the French 'Ministère de l'Enseignement Supérieur et de la Recherche'.

## Additional information

### Funding

| Funder | Grant reference number | Author |
| --- | --- | --- |
| Agence Nationale de la Recherche | ANR-13-BV5-0014-01 | Romuald Nargeot |
| Agence Nationale de la Recherche | ANR-10-Idex-03-02 | Alexis Bédécarrats |

The funders had no role in study design, data collection and interpretation, or the decision to submit the work for publication.

### Author contributions

Alexis Bédécarrats, Data curation, Formal analysis, Funding acquisition, Investigation, Methodology, Writing - review and editing; Laura Puygrenier, Data curation, Formal analysis, Funding acquisition, Investigation, Writing - review and editing; John Castro O'Byrne, Formal analysis, Investigation; Quentin Lade, Investigation; John Simmers, Conceptualization, Supervision, Funding acquisition, Validation, Writing - original draft, Writing - review and editing; Romuald Nargeot, Conceptualization,

Formal analysis, Supervision, Funding acquisition, Investigation, Methodology, Writing - original draft, Project administration, Writing - review and editing

### Author ORCIDs
Alexis Bédécarrats http://orcid.org/0000-0003-3621-5639
John Simmers http://orcid.org/0000-0002-7487-4638
Romuald Nargeot https://orcid.org/0000-0002-7939-0333

### Decision letter and Author response
Decision letter https://doi.org/10.7554/eLife.68651.sa1
Author response https://doi.org/10.7554/eLife.68651.sa2

## Additional files

### Supplementary files
- Source code 1. Phase-plan plot.

- Transparent reporting form

### Data availability
Source data file have been provided for Figures 2,3,10: Bédécarrats, Alexis et al. (2021), Organelle calcium-derived voltage oscillations in pacemaker neurons drive the motor program for food-seeking behavior in Aplysia, Dryad, Dataset, https://doi.org/10.5061/dryad.pvmcvdnkr.

The following dataset was generated:

| Author(s) | Year | Dataset title | Dataset URL | Database and Identifier |
|---|---|---|---|---|
| Bédécarrats A, Puygrenier L, Castro O'Byrne J, Lade Q, Simmers J, Nargeot R | 2021 | Data from: Organelle calcium-derived voltage oscillations in pacemaker neurons drive the motor program for food-seeking behavior in Aplysia | https://doi.org/10.5061/dryad.pvmcvdnkr | Dryad Digital Repository, 10.5061/dryad.pvmcvdnkr |

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
