## [Decision Letter]

**Acceptance summary:**

This paper provides evidence for a novel mechanism that helps explain the irregularities of action potential bursts underlying the biting motor program in *Aplysia*. The authors justified their conclusion that calcium release arising from intracellular organelles underlies this voltage oscillations. They addressed the concern that extra-neuronal calcium might play a role.

**Decision letter after peer review:**

Thank you for submitting your article "Organelle calcium-derived voltage oscillations in pacemaker neurons drive food-seeking behavior in *Aplysia*" for consideration by *eLife*. Your article has been reviewed by 3 peer reviewers, including Paul Katz as the Reviewing Editor and Reviewer #1, and the evaluation has been overseen by Ronald Calabrese as the Senior Editor. The following individual involved in review of your submission has agreed to reveal their identity: Scott Hooper (Reviewer #2).

Essential revisions :

The paper was in general very well received by the reviewers who were impressed by the novelty of the mechanism uncovered as a driver of a fictive motor program and thus likely behavior. The reviewers provide useful feedback for revision in the Recommendations for the Authors. There are two major points that should be thoroughly addressed.

1. A main concern was the question of where the plateau potentials that drive network activity are being generated. The references that are cited don't support the claims made about B63. It is however possible that the authors can present other data or cite other work to support their claim. Even if they can't, there is little doubt that activity in B63 is important for initiating activity in the feeding CPG. As a result, what drives the phasic depolarization in B63 is also of interest (even if the network driving plateau potentials are subsequently generated in B31/32). See Major comment #1 from Reviewer 3.

2. Please remove the Choline for Na^+^ substitution experiments from the manuscript, See major comment #6 of Reviewer 2.

*Reviewer #1 (Recommendations for the authors):*

The labeling of the traces was inconsistent. For example, in Figure 6A, B63r and B63l are above their respective traces, but in Figure 6B, the labels are below their traces. A standard position of the label either above, below, or preferably to the left of the trace should be adopted.

In the electrical coupling circuits, it is common to show a resistor symbol to indicate electrical coupling. The single diagonal swipe is not a standard representation. Also, just for simplicity, put B63r to the right of B63l in A and in the plot in D.

Figure 6C shows an extracellular recording of B8r, which is not sufficient to demonstrate an absence of subthreshold oscillations.

Figure 7C, Phase lag cannot be represented in seconds. Phase is unit-less. Seconds would indicate latency.

*Reviewer #2 (Recommendations for the authors):*

1) With respect to the non-voltage dependence of the small oscillation, note more strongly that cell membrane potential, changed either by current injection or by spontaneous plateau potentials, does not re-set the oscillation. Refer to Figure 6A, Suppl. Figure 1, and Suppl. Figure 4 in doing so.

2) In Figure 5 the authors demonstrate that mean small oscillation amplitude varies with membrane potential, becoming smaller with depolarization and larger with hyperpolarization. This suggests that the cell membrane channels opened by the intracellular mechanism carry Na or Ca. However, the authors do not provide quantitative measurements and summaries of the changes in this amplitude. Please do so as is so admirably done for the other types of data in the manuscript.

3) I do not understand the logic of lines 315-324 in the manuscript, the portion arguing that the fact that the B63 small oscillation cycle is out of phase with that of other buccal CPG neurons, e.g., B31 (the text associated with Figure 7), indicates the oscillation is not an emergent network property. That two parts of an network oscillate out of phase is not, to my mind, evidence that the oscillations are not an emergent network property. A classic counter-example is half-center oscillation, which in some instantiations generates 180º out-of-phase oscillations in the two half-centers, but in which the oscillation is clearly a network-emergent property. That the oscillations originate in the B63 cell pair, and are passively transmitted to the other neurons is instead best demonstrated by the data in Figure 10. I would move this text to this figure. As to some of the (presumably) follower neurons being out of phase, this could be explained by the delays in cell-to-cell communication the authors present in the Discussion. It is unfortunate that, in the heparin experiments, the cell pairs never included recordings from the out-of-phase neurons, as such recordings would resolve whether these out-of-phase oscillations depended on B63 oscillations.

4) In the low Ca-cobalt saline used to block synaptic transmission, B63 plateau duration dramatically increases. My rough measurements from Figure 1, 2, and the first plateau in Figure 3A show plateau durations in the 10 to 40 s range. The initial low Ca-cobalt saline plateau potentials in Figure 3A are again in this duration range, but after this saline had been on for a longer time, plateau potential durations became longer (40 to 60 s in Figure 4, 1 to 2.5 min in Figure 3B, 3 to 4 min in Figure 5, and 5 min in Figure 6A). I do not believe that this large change in any way affects manuscript conclusions, but it should be pointed out as it may be of interest to workers in the membrane conductances underlying plateau potential production.

5) I do not understand the comparisons being made in lines 308-314 in the manuscript, the comparison being made between different pairs, e.g., B31//B31 vs. B31/B63. Please expand or clarify what is being compared here.

6) In the experiments in which extracellular Na and Ca concentrations are being changed, all the data in which Na was replaced by choline must be stricken from the manuscript. Thuma and Hooper [Thuma, JB, Hooper, SL (2018) Choline and NMDG directly reduce outward currents: reduced outward current when these substances replace Na^+^ is alone not evidence of Na+-activated K^+^ currents. J Neurophysiol 120:3217-3233. doi: 10.1152/jn.00871.2017) have shown that choline alone blocks K channels, as would be expected for tertiary or quaternary amines. In the opinion of this reviewer, the prior long use of this substitution without testing for the effects of simple choline addition is inexplicable and a dismaying example of how error can remain in science for long periods. Regardless, this error must not continue now that these direct effects have been described. In the case at hand, I agree that the effects are indeed due to the lack of Na, but the use of choline actually diminishes the intended demonstration, as the disappearance of the oscillation in choline could be due to either reduced Na or K currents. The disappearance in TTX is sufficient and unambiguous.

7) In the Discussion, I would add a paragraph about how B63 membrane potentials do not re-set the underlying oscillation. This disconnect from neuron activity is striking. One thinks of neurons as being largely electrical entities. For a process that is itself altering neuron electrical activity to not be affected by that activity seems worth commenting on.

8) Lines 542-545 in the present manuscript about cytosolic Ca buffering expressed a concern I had throughout the manuscript, the idea that Ca release from ER and mitochondria would alter global cytoplasm [Ca]. I would expect this Ca would instead form a shell around the ER and mitochondria membranes, just as Ca entry across the cell membrane forms a shell of high [Ca] under the membrane. Admittedly, in muscles Ca release from the sarcoplasmic reticulum causes a global change in muscle cell cytoplasm [Ca], but that is a highly evolved system in which that is the goal. Do the authors imagine perhaps that there is specialized ER or mitochondria located close enough to the cell membrane that Ca moves to form a shell under the cell membrane? That Ca released from the ER and mitochondria diffuses to form a shell under the cell membrane? Or that the cytoplasmic buffering system is overwhelmed? Of course, injecting positive current through an electrode with K or Na ions does depolarize the membrane, and it is presumably not the injected ions that are depolarizing the cell membrane, rather a redistribution of charge in the cytoplasm to maintain iso-voltage in it that results in the ion shell under the cell membrane being altered so as to depolarize the membrane. But these ions are not highly buffered, and I do not know the literature of whether injecting Ca through an electrode into a cell depolarizes it. Regardless, this difficulty of release of Ca from organelles embedded in a cytoplasm with a high Ca buffering capacity causing a change in potential across the cell membrane should be more fully dealt with in the Discussion. I think it is exactly the same problem they mention with the B63 oscillation being transmitted to electrically coupled cells in this part of the Discussion. Including references showing that Ca injection into a cell through an electrode causes the cell to depolarize would be sufficient.

[Editors' note: further revisions were suggested prior to acceptance, as described below.]

Thank you for submitting your article "Organelle calcium-derived voltage oscillations in pacemaker neurons drive the motor program for food-seeking behavior in *Aplysia*" for consideration by *eLife*. Your article has been reviewed by 3 peer reviewers, including Paul Katz as the Reviewing Editor and Reviewer #1, and the evaluation has been overseen by Ronald Calabrese as the Senior Editor. The following individual involved in review of your submission has agreed to reveal their identity: Scott Hooper (Reviewer #2).

Essential revisions:

After consultation with the reviewers, it was decided that the authors should respond in the paper to the interstitial cell hypothesis that has been raised by Reviewer 1. We are all sorry that this was raised at a late stage in the review process, but we agree that it is better that the paper address the issue than to be proven wrong later. If the mechanism turns out to be a novel involvement of the interstitial cells, then the impact of the paper would be increased.

We debated about whether a mechanism involving changes in extracellular calcium concentration could account for low amplitude oscillations observed in B63. Ultimately, we decided that it was up to the authors to decide whether their data are consistent with two alternate hypotheses (release and sequestration of calcium intracellularly by organelles or extracellularly by interstitial cells) or whether they can eliminate one of these hypotheses. One consideration is whether the authors can determine the resting calcium conductance and can estimate the change in extracellular calcium concentration that would be needed to observe the small membrane potential fluctuations based on changes to the predicted calcium equilibrium potential. It would be advisable to consider looking at persistent sodium current magnitudes to get an idea of how large gCa would need to be to generate sufficient calcium current.

The interstitial cell hypothesis explains a concern that one reviewer had expressed, and that the authors now explicitly cover in the manuscript (lines 237-278), about how changing B63's membrane potential does not change the underlying oscillation in B63. If the source of the underlying oscillation were the interstitial cells, current injection into B63 would not be expected to change the oscillation, just whether the oscillation were able to trigger a burst, exactly what the data show.

*Reviewer #1 (Recommendations for the authors):*

The Central Conclusion as expressed in the abstract is that there is a "rhythmic signal, which is endogenous and specific to the B63 cells, [and] is generated by organelle-derived intracellular calcium fluxes that activate voltage-independent plasma membrane channels."

The evidence that the oscillations derive from organelle calcium fluxes is indirect. It is possible that the calcium oscillations are arising from non-neural interstitial cells that have been shown to take up and release calcium. I recently became aware of work by Ghislain Nicaise on gliointerstitial cells in molluscs: Nicaise, Ghislain. "The gliointerstitial system of molluscs." International Review of Cytology 34 (1973): 251-332.

Here are some additional citations that show that interstitial cells concentrate and release calcium from vesicles directly opposed to neurons.

1: Keicher E, Maggio K, Hernandez-Nicaise ML, Nicaise G. The abundance of *Aplysia* gliagrana depends on Ca^2+^ and/or Na^+^ concentrations in sea water. Glia. 1992;5(2):131-8. doi: 10.1002/glia.440050207. PMID: 1533611.

2: Maggio K, Watrin A, Keicher E, Nicaise G, Hernandez-Nicaise ML. Ca(2+)-ATPase and Mg(2+)-ATPase in *Aplysia* glial and interstitial cells: an EM cytochemical study. J Histochem Cytochem. 1991 Dec;39(12):1645-58. doi: 10.1177/39.12.1719071. PMID: 1719071.

3: Keicher E, Bilbaut A, Maggio K, Hernandez-Nicaise ML, Nicaise G. The Desheathed Periphery of *Aplysia* Giant Neuron. Fine Structure and Measurement of [Ca^2+^]o Fluctuations with Calcium-selective Microelectrodes. Eur J Neurosci. 1991 Oct;3(1):10-17. doi: 10.1111/j.1460-9568.1991.tb00806.x. PMID: 12106264.

4: Keicher E, Maggio K, Hernandez-Nicaise ML, Nicaise G. The lacunar glial zone at the periphery of *Aplysia* giant neuron: volume of extracellular space and total calcium content of gliagrana. Neuroscience. 1991;42(2):593-601. doi: 10.1016/0306-4522(91)90401-9. PMID: 1896135.

The manipulations of calcium that the authors of the current submission performed are consistent with an alternate hypothesis, namely that the calcium-dependent oscillations are not derived from intracellular organelles in neuron B63, but rather from non-neural cells that abut B63.

There may be ways to disprove this hypothesis, but unless the authors address them, they need to alter their title and conclusion.

*Reviewer #2 (Recommendations for the authors):*

Reviewer 1 is satisfied on his points.

*Reviewer #3 (Recommendations for the authors):*

My previous comments have all been addressed.

---

## [Author Response]

Essential revisions:The paper was in general very well received by the reviewers who were impressed by the novelty of the mechanism uncovered as a driver of a fictive motor program and thus likely behavior. The reviewers provide useful feedback for revision in the Recommendations for the Authors. There are two major points that should be thoroughly addressed.1. A main concern was the question of where the plateau potentials that drive network activity are being generated. The references that are cited don't support the claims made about B63. It is however possible that the authors can present other data or cite other work to support their claim. Even if they can't, there is little doubt that activity in B63 is important for initiating activity in the feeding CPG. As a result, what drives the phasic depolarization in B63 is also of interest (even if the network driving plateau potentials are subsequently generated in B31/32). See Major comment #1 from Reviewer 3.2. Please remove the Choline for Na^+^ substitution experiments from the manuscript, see major comment #6 of Reviewer 2.Reviewer #1 (Recommendations for the authors):The labeling of the traces was inconsistent. For example, in Figure 6A, B63r and B63l are above their respective traces, but in Figure 6B, the labels are below their traces. A standard position of the label either above, below, or preferably to the left of the trace should be adopted.

As requested by the reviewer, the neuron labels in Figure 6 have been moved to the left and above the traces.

In the electrical coupling circuits, it is common to show a resistor symbol to indicate electrical coupling. The single diagonal swipe is not a standard representation. Also, just for simplicity, put B63r to the right of B63l in A and in the plot in D.

In Figure 6 (and Figure 1B), resistor symbols now indicate electrical coupling as suggested by the reviewer.

Also, B63l has been placed to the left of B63r in A, and the data from left cells are now reported on the left side of the relevant plots in D.

Figure 6C shows an extracellular recording of B8r, which is not sufficient to demonstrate an absence of subthreshold oscillations.

Figure 6C, to better indicate that the B8 trace is in fact an intracellular and not an extracellular recording, in the revised figure version, the action potentials in the B8 trace have been less truncated and the figure legend (Line 1160) now reads “Simultaneous intracellular recordings from a B63 cell and a non-coupled contralateral B8 motor neuron”

Figure 7C, Phase lag cannot be represented in seconds. Phase is unit-less. Seconds would indicate latency.

Figure 7C: As rightly requested, “Phase lag” has been replaced by “Latency”

Reviewer #2 (Recommendations for the authors):1) With respect to the non-voltage dependence of the small oscillation, note more strongly that cell membrane potential, changed either by current injection or by spontaneous plateau potentials, does not re-set the oscillation. Refer to Figure 6A, Suppl. Figure 1, and Suppl. Figure 4 in doing so.

As suggested by the reviewer, the revised manuscript now states more strongly that neither spontaneous plateau potentials nor experimental depolarization/hyperpolarization of the B63 cell were able to modify or reset the oscillation as illustrated in Figure 6A, Figure 3—figure supplement 1 and Figure 6—figure supplement 1: Lines 253-257, “They also indicated that a change in B63’s membrane potential, either in response to experimental manipulation or during the plateau potentials themselves, neither changed the small oscillation’s cycle period nor caused phase-resetting (see also Figure 6A, Figure 3—figure supplement 1, Figure 6—figure supplement 1A,B).”

2) In Figure 5 the authors demonstrate that mean small oscillation amplitude varies with membrane potential, becoming smaller with depolarization and larger with hyperpolarization. This suggests that the cell membrane channels opened by the intracellular mechanism carry Na or Ca. However, the authors do not provide quantitative measurements and summaries of the changes in this amplitude. Please do so as is so admirably done for the other types of data in the manuscript.

A quantification of mean amplitude of the small oscillation at the different membrane potentials is now reported in Figure 5C,E and the legend. Reference to these new figure panels have also been added in the Results section (line 267). Note that in the revised Figure 5C, the data obtained at -80mV and -70mV are now plotted on the left and right sides, respectively, of each graph (rather than the inverse as in the original Figure).

3) I do not understand the logic of lines 315-324 in the manuscript, the portion arguing that the fact that the B63 small oscillation cycle is out of phase with that of other buccal CPG neurons, e.g., B31 (the text associated with Figure 7), indicates the oscillation is not an emergent network property. That two parts of an network oscillate out of phase is not, to my mind, evidence that the oscillations are not an emergent network property. A classic counter-example is half-center oscillation, which in some instantiations generates 180º out-of-phase oscillations in the two half-centers, but in which the oscillation is clearly a network-emergent property. That the oscillations originate in the B63 cell pair, and are passively transmitted to the other neurons is instead best demonstrated by the data in Figure 10. I would move this text to this figure. As to some of the (presumably) follower neurons being out of phase, this could be explained by the delays in cell-to-cell communication the authors present in the Discussion. It is unfortunate that, in the heparin experiments, the cell pairs never included recordings from the out-of-phase neurons, as such recordings would resolve whether these out-of-phase oscillations depended on B63 oscillations.

We accept that the logic here was confusing, especially with the use of terminology such as ‘emergent property of a network’. Lines 320-324 of the original manuscript stating that “the relative timing and amplitude differences therefore support the conclusion that the voltage oscillation is not an equivalent emergent property of a network of electrically-coupled neurons” have been removed in the revised manuscript.

Moreover, as suggested by the reviewer, the conclusion of the heparin experiments (Lines 415-419) is now used to better make these points and states: “these results, along with the previous findings of a consistently larger amplitude and phase advance of B63’s voltage oscillation (see Figure 6), further indicated that it originates in, and is specific to, the B63 cell pair – the sole necessary and sufficient elements for triggering BMPs – and spreads presumably via gap junctional connections throughout the remaining circuit.”

4) In the low Ca-cobalt saline used to block synaptic transmission, B63 plateau duration dramatically increases. My rough measurements from Figure 1, 2, and the first plateau in Figure 3A show plateau durations in the 10 to 40 s range. The initial low Ca-cobalt saline plateau potentials in Figure 3A are again in this duration range, but after this saline had been on for a longer time, plateau potential durations became longer (40 to 60 s in Figure 4, 1 to 2.5 min in Figure 3B, 3 to 4 min in Figure 5, and 5 min in Figure 6A). I do not believe that this large change in any way affects manuscript conclusions, but it should be pointed out as it may be of interest to workers in the membrane conductances underlying plateau potential production.

Inspection of our recordings does not indicate that plateau potential durations increased progressively with time under application of low Ca+Co saline. However, there was often a relatively abrupt increase in plateau durations soon after perfusion onset due to the removal of synaptic inhibition from retractor generator neurons, which contributes to terminating B63 plateaus during BMP genesis. This point is now added to the revised text (Lines 181185): ”It also is noteworthy that in Low Ca+Co saline, due to the resultant suppression of burst-terminating inhibitory synaptic input from buccal neurons of the retraction generator (see Figure 1B), B63’s plateau potentials still had variable durations, but overall lasted longer than in normal saline (compare Figure 2A and Figure 6—figure supplement 1A)”.

5) I do not understand the comparisons being made in lines 308-314 in the manuscript, the comparison being made between different pairs, e.g., B31//B31 vs. B31/B63. Please expand or clarify what is being compared here.

The reason for quantifying and comparing oscillation phase delays between different neuron pairs (see Figure 7; Lines 308-314 of the original manuscript) was to gain indirect access to the relative timing between pairs of cells that were not recorded simultaneously. For example, the B31/B30 neurons were not pair-wise recorded. But, the absence of a statistical difference in the phase delays between the recorded pairs B31/B63 and B30/B63 allowed deducing that the oscillations of B31 and B30 were in fact synchronous. However, since such comparisons provide only indirect evidence and manifestly confuse rather than enlighten, they have been removed from the revised manuscript.

6) In the experiments in which extracellular Na and Ca concentrations are being changed, all the data in which Na was replaced by choline must be stricken from the manuscript. Thuma and Hooper [Thuma, JB, Hooper, SL (2018) Choline and NMDG directly reduce outward currents: reduced outward current when these substances replace Na^+^ is alone not evidence of Na+-activated K^+^ currents. J Neurophysiol 120:3217-3233. doi: 10.1152/jn.00871.2017) have shown that choline alone blocks K channels, as would be expected for tertiary or quaternary amines. In the opinion of this reviewer, the prior long use of this substitution without testing for the effects of simple choline addition is inexplicable and a dismaying example of how error can remain in science for long periods. Regardless, this error must not continue now that these direct effects have been described. In the case at hand, I agree that the effects are indeed due to the lack of Na, but the use of choline actually diminishes the intended demonstration, as the disappearance of the oscillation in choline could be due to either reduced Na or K currents. The disappearance in TTX is sufficient and unambiguous.

We thank the reviewer for passionately pointing out the unintended error of our ways with the use of choline for Na substitution. All mention and data related to the use a of 0 sodium solution containing choline have been removed from the revised manuscript (main text, Materials and methods ‘Modified salines’ section and Figure 8). Accordingly, statistical treatment of the data has been rerun with the Mann-Whitney test for comparing two independent groups (rather than the Kruskal-Wallis and post-hoc Conover tests for comparing three groups). The ‘Statistical analysis’ section in the Materials and methods section has also been modified accordingly (Lines 710-711).

7) In the Discussion, I would add a paragraph about how B63 membrane potentials do not re-set the underlying oscillation. This disconnect from neuron activity is striking. One thinks of neurons as being largely electrical entities. For a process that is itself altering neuron electrical activity to not be affected by that activity seems worth commenting on.

A new paragraph has now been added to the Discussion (Lines 485-500) in which we offer an explanation for the inability of plateau potentials or experimental manipulation of the somatic membrane potential to modify the small oscillation’s cycle period or cause resetting. This paragraph, which also relates to the following reviewer comment 8 now states: “The organelle calcium release and resulting voltage oscillation could be localized to a neuronal compartment, such as the junctional synapses or neuropile where a close proximity between ER, mitochondria, and plasma membrane is likely to exist (Thompson et al., 1976), but which is remote from the soma compartment where plateau potentials occur. In this scheme, local calcium release and IP3/calcium-dependent amplification/propagation along the ER membrane could activate nearby plasma membrane conductances at this soma distant site and generate a voltage oscillation that is passively backpropagated to the cell body where it can trigger plateau potentials. Such spatially separate loci for intracellular calcium oscillation and plateau potential production in turn provides a plausible explanation for several of our experimental observations, including: (1) the ability of local organelle calcium release to produce a membrane potential oscillation of B63, despite the presumed presence of strong cytoplasmic calcium buffering mechanisms; (2) the near synchronous expression of the voltage oscillation, but not the somatic plateau potentials, throughout B63’s gap junction-coupled cell partners; and (3) the inability of experimental soma depolarization or spontaneous plateau potentials and resulting somatic calcium influxes to modify or reset the remotely-produced calcium oscillation.”

8) Lines 542-545 in the present manuscript about cytosolic Ca buffering expressed a concern I had throughout the manuscript, the idea that Ca release from ER and mitochondria would alter global cytoplasm [Ca]. I would expect this Ca would instead form a shell around the ER and mitochondria membranes, just as Ca entry across the cell membrane forms a shell of high [Ca] under the membrane. Admittedly, in muscles Ca release from the sarcoplasmic reticulum causes a global change in muscle cell cytoplasm [Ca], but that is a highly evolved system in which that is the goal. Do the authors imagine perhaps that there is specialized ER or mitochondria located close enough to the cell membrane that Ca moves to form a shell under the cell membrane? That Ca released from the ER and mitochondria diffuses to form a shell under the cell membrane? Or that the cytoplasmic buffering system is overwhelmed? Of course, injecting positive current through an electrode with K or Na ions does depolarize the membrane, and it is presumably not the injected ions that are depolarizing the cell membrane, rather a redistribution of charge in the cytoplasm to maintain iso-voltage in it that results in the ion shell under the cell membrane being altered so as to depolarize the membrane. But these ions are not highly buffered, and I do not know the literature of whether injecting Ca through an electrode into a cell depolarizes it. Regardless, this difficulty of release of Ca from organelles embedded in a cytoplasm with a high Ca buffering capacity causing a change in potential across the cell membrane should be more fully dealt with in the Discussion. I think it is exactly the same problem they mention with the B63 oscillation being transmitted to electrically coupled cells in this part of the Discussion. Including references showing that Ca injection into a cell through an electrode causes the cell to depolarize would be sufficient.

We do not wish to suggest that the voltage oscillation results from a global cytoplasmic change in calcium concentration. As in many other cells, the calcium oscillation may be generated locally (e.g. in the neuropile or junctional synapses) where a close apposition between ER, mitochondria and plasma membrane may occur. At this local site the calcium oscillation may be translated into a voltage oscillation that backpropagates to the soma where plateaus are produced. As indicated in our response to preceding point 7, this hypothesis is now developed in Lines 485-500 of the revised manuscript.

[Editors' note: further revisions were suggested prior to acceptance, as described below.]

Essential revisions:After consultation with the reviewers, it was decided that the authors should respond in the paper to the interstitial cell hypothesis that has been raised by Reviewer 1. We are all sorry that this was raised at a late stage in the review process, but we agree that it is better that the paper address the issue than to be proven wrong later. If the mechanism turns out to be a novel involvement of the interstitial cells, then the impact of the paper would be increased.We debated about whether a mechanism involving changes in extracellular calcium concentration could account for low amplitude oscillations observed in B63. Ultimately, we decided that it was up to the authors to decide whether their data are consistent with two alternate hypotheses (release and sequestration of calcium intracellularly by organelles or extracellularly by interstitial cells) or whether they can eliminate one of these hypotheses. One consideration is whether the authors can determine the resting calcium conductance and can estimate the change in extracellular calcium concentration that would be needed to observe the small membrane potential fluctuations based on changes to the predicted calcium equilibrium potential. It would be advisable to consider looking at persistent sodium current magnitudes to get an idea of how large gCa would need to be to generate sufficient calcium current.The interstitial cell hypothesis explains a concern that one reviewer had expressed, and that the authors now explicitly cover in the manuscript (lines 237-278), about how changing B63's membrane potential does not change the underlying oscillation in B63. If the source of the underlying oscillation were the interstitial cells, current injection into B63 would not be expected to change the oscillation, just whether the oscillation were able to trigger a burst, exactly what the data show.Reviewer #1 (Recommendations for the authors):The Central Conclusion as expressed in the abstract is that there is a "rhythmic signal, which is endogenous and specific to the B63 cells, [and] is generated by organelle-derived intracellular calcium fluxes that activate voltage-independent plasma membrane channels."The evidence that the oscillations derive from organelle calcium fluxes is indirect. It is possible that the calcium oscillations are arising from non-neural interstitial cells that have been shown to take up and release calcium. I recently became aware of work by Ghislain Nicaise on gliointerstitial cells in molluscs: Nicaise, Ghislain. "The gliointerstitial system of molluscs." International Review of Cytology 34 (1973): 251-332.Here are some additional citations that show that interstitial cells concentrate and release calcium from vesicles directly opposed to neurons.1: Keicher E, Maggio K, Hernandez-Nicaise ML, Nicaise G. The abundance of Aplysia gliagrana depends on Ca^2+^ and/or Na^+^ concentrations in sea water. Glia. 1992;5(2):131-8. doi: 10.1002/glia.440050207. PMID: 1533611.2: Maggio K, Watrin A, Keicher E, Nicaise G, Hernandez-Nicaise ML. Ca(2+)-ATPase and Mg(2+)-ATPase in Aplysia glial and interstitial cells: an EM cytochemical study. J Histochem Cytochem. 1991 Dec;39(12):1645-58. doi: 10.1177/39.12.1719071. PMID: 1719071.3: Keicher E, Bilbaut A, Maggio K, Hernandez-Nicaise ML, Nicaise G. The Desheathed Periphery of Aplysia Giant Neuron. Fine Structure and Measurement of [Ca^2+^]o Fluctuations with Calcium-selective Microelectrodes. Eur J Neurosci. 1991 Oct;3(1):10-17. doi: 10.1111/j.1460-9568.1991.tb00806.x. PMID: 12106264.4: Keicher E, Maggio K, Hernandez-Nicaise ML, Nicaise G. The lacunar glial zone at the periphery of Aplysia giant neuron: volume of extracellular space and total calcium content of gliagrana. Neuroscience. 1991;42(2):593-601. doi: 10.1016/0306-4522(91)90401-9. PMID: 1896135.The manipulations of calcium that the authors of the current submission performed are consistent with an alternate hypothesis, namely that the calcium-dependent oscillations are not derived from intracellular organelles in neuron B63, but rather from non-neural cells that abut B63.There may be ways to disprove this hypothesis, but unless the authors address them, they need to alter their title and conclusion.

Reviewer #1 has raised an interesting point as to whether non-neural glial cells and their extracellular regulation of the ionic environment could play a role in the genesis of the B63 neuron’s voltage oscillation.

Nevertheless, our major arguments for a neuronal, rather than a glial cell role in B63’s oscillation genesis are: (1) its suppression in B63 (and other coupled neurons) by the intracellular injection of heparin uniquely into B63; (2) manipulation of B63’s membrane potential and the resulting changes in oscillation magnitude, indicating that the oscillation is produced by conductances located in the cell’s membrane and is not an ephaptic consequence of changes in the surrounding ionic environment; (3) the absence of a voltage-sensitivity of the oscillation’s occurrence, thus discounting a contribution of voltage-dependent and extracellular calcium-sensitive channels, such as those conveying INaP; (4) oscillation genesis arising spontaneously and at a constant and restricted location (B63) with inter-neuronal variability in oscillation amplitudes and latencies also not readily complying with a functional contribution of a glial syncytium. It is also significant that this specific cellular locus at which the oscillation arises is precisely the only network neurons that are necessary and sufficient for triggering buccal motor pattern genesis.

Despite these arguments, a contribution of a neuron-glia interplay in the genesis and/or propagation of the oscillation is still not excluded. However, we strongly believe that our data clearly indicate that the low amplitude voltage oscillation, at least in large part, derives from intracellular organelle calcium fluxes in neuron B63.

In the revised manuscript, as indicated in the accompanying document with traceable corrections, several parts of the Discussion have been rewritten to (1) present the hypothesis of a glial origin for the oscillation, (2) more explicitly report the arguments for a neuronal origin, without excluding a possible parallel regulatory role for a glial syncytium. We have also been more circumspect in some of the wording about the underlying origin of B63’s voltage oscillation in the Results section, eight new bibliographical references relative to glial cell function, including one suggested by the reviewer, have now been added to the revised manuscript.